# PHEIGES: all-cell-free phage synthesis and selection from engineered genomes

Antoine Levrier[1,2], Ioannis Karpathakis[1,3], Bruce Nash[4], Steven D. Bowden[5], Ariel B. Lindner [2] ✉ & Vincent Noireaux [1] ✉

Bacteriophages constitute an invaluable biological reservoir for biotechnology and medicine. The ability to exploit such vast resources is hampered by the lack of methods to rapidly engineer, assemble, package genomes, and select phages. Cell-free transcription-translation (TXTL) offers experimental settings to address such a limitation. Here, we describe PHage Engineering by In vitro Gene Expression and Selection (PHEIGES) using T7 phage genome and Escherichia coli TXTL. Phage genomes are assembled in vitro from PCR-amplified fragments and directly expressed in batch TXTL reactions to produce up to $10^{11}$ PFU/ml engineered phages within one day. We further demonstrate a significant genotype-phenotype linkage of phage assembly in bulk TXTL. This enables rapid selection of phages with altered rough lipopolysaccharides specificity from phage genomes incorporating tail fiber mutant libraries. We establish the scalability of PHEIGES by one pot assembly of such mutants with fluorescent gene integration and 10% length-reduced genome.

Bacteriophages (phages) comprise an immense reservoir of biotechnologically-relevant bioactive materials such as DNA engineering tools[1] and phage display[2] with broad applications in phage therapy[3–8], nanotechnology[9,10], and vaccine scaffolds engineering[11,12]. Despite this, current in vivo phage engineering approaches, based on CRISPR[13–15], yeast assembly[16,17] or integrases[18] limit phage's usage due to time consuming cloning steps and complex selection processes[19], especially challenging for obligated lytic phages[20]. On the other hand, in vitro genome engineering is emerging as a complementary approach to assemble and edit bacteriophages with novel properties. In vitro (Gibson[21] or Golden gate[22]) assemblies of bacteriophage genomes from PCR fragments were expressed in bacteria[22], packaged in pro-capsids[23], or rebooted in cell-free transcription-translation (TXTL)[21]. Despite their simplicity, cellular transformation and amplification induce bias and complexify downstream selection processes while genome packaging in procapsid limits the variety of genomes to engineer. Thus, TXTL offers an alternative experimental setting to achieve phage engineering and synthesis, complementary to and potentially faster than existing methods[24,25]. TXTL provides an unparalleled speed to design, build and test DNAs[26]. TXTL has proven effective for prototyping gene circuits[27–30], manufacturing biologics[31], or building biological systems from the ground up[32–34]. As it gains in strength and versatility[25,35,36], TXTL can be challenged for executing larger DNAs. In this regard, the cell-free synthesis (CFS) of infectious T7 and T4 phages in an E. coli TXTL system demonstrates that natural genomic DNAs encoding for tens of genes can be achieved in vitro[24,37,38]. The CFS of phages reduces the use of bacteria, pathogens in particular, and provides a boundary-free environment meant scalable and un-biased expression of large and complex genetic circuits. Additionally E. coli TXTL does not limit the applications to coliphages as it supports the synthesis of phages targeting other bacteria when specific transcription factors are expressed[39]. Yet, currently, purification of intact phage genomes and their engineering is time-consuming and challenging[24] due to the size (> 40kbp) and fragility of the genomic DNA molecules while current DNA assembly methods do not interface well with TXTL, require purification steps[40] limit phage titers[21] and hence the library size that can be achieved. The latter depends on cellular passage for correct genotype-phenotype linkage and amplification, further limiting the process.

[1]School of Physics and Astronomy, University of Minnesota, Minneapolis, MN 55455, USA. [2]Université Paris Cité, INSERM U1284, Center for Research and Interdisciplinarity, F-75006 Paris, France. [3]Facultatea de Biotehnologii, USAMV Bucuresti, Sector 1, Cod 011464 Bucureşti, Romania. [4]DNA Learning Center, Cold Spring Harbor Laboratory, Cold Spring Harbor, NY 11724, USA. [5]Department of Food Science and Nutrition, University of Minnesota, Saint Paul, MN 55108, USA. ✉e-mail: ariel.lindner@inserm.fr; noireaux@umn.edu

In this work, we address these critical bottlenecks with PHEIGES (PHage Engineering by In vitro Gene Expression and Selection)—a rapid all-cell-free workflow to achieve phage genome engineering, synthesis, and selection. PHEIGES couples a recent DNA assembly procedure[23] with an all-*E. coli* TXTL toolbox[25] to synthesize and select engineered T7 phages in under one day (Fig. 1). PHEIGES enables basic biology probing of phage packaging and phage-host interactions and can be used iteratively to rapidly perform permissive phage engineering and selection, without the need of cellular amplification or other synthetic means for genotype-phenotype coupling, such as cell-sized emulsion droplets.

## Results

### Cell-free transcription-translation

The myTXTL system[25,41] uses the endogenous *E. coli* core RNA polymerase and sigma factor 70 present in the lysate as the sole primary transcription proteins. This system does not contain any remaining live *E. coli* cells (Supplementary Fig. 1). Genes are expressed either from plasmids or linear dsDNA. In this work, all the TXTL reactions were carried out in batch mode at the scale of 1–10 μl, either in 1.5 mL tubes or in well plates. In batch mode, 80–100 μM of deGFP protein are produced after 12 h of incubation (Supplementary Fig. 2) from an *E. coli* promoter (P70a[25]). With the bacteriophage T7 promoter, 100–120 μM deGFP proteins are produced in 3–6 h. The wild-type phage T7 (T7 WT) is synthesized from its genome at a concentration of $10^{10}$–$10^{11}$ PFU/ml (plaque-forming units per milliliter) after 3 h of incubation (Supplementary Fig. 3).

### PHEIGES workflow

Our workflow design was based on specific requirements: (i) enable gene addition, deletion, and mutation concurrently at any position in a single DNA assembly reaction, (ii) DNA assembly should be cell-free, (iii) DNA assembly should interface seamlessly with TXTL (Fig. 1), (iv) low-cost, and (v) easy to handle and technically accessible from start to end. These conditions eliminate many of the DNA assembly methods that have been reported recently. Editing procedures based on CRISPR[13,15,14,42] or living chassis[16,43] are time-consuming and relatively costly. The Gibson and Golden Gate DNA assembly methods are achieved in buffers that do not interface well with TXTL, which requires additional purification steps after assembly[21]. Other methods, such as SLiCE[44], also do not interface well with TXTL as the ligation reaction inhibits cell-free gene expression. Our attention focused on a recent method to assemble and package phage genomes into procapsids[23].

The T7 genome is re-assembled from long PCR fragments (<12 kbp) with overlapping sequences using a cheap assembly mix containing only an exonuclease[23], followed by heat inactivation of the enzyme. Annealed fragments are directly expressed in TXTL without additional steps enabling the synthesis and selection of T7 phage

variants that integrate gene addition, deletion, mutation. The workflow, achieved in under one day, delivers phages at titers comparable to titers obtained from bacterial lysate ($10^{10-11}$ PFU/ml) (Fig. 1).

### Plasmid construction and T7 WT rebooting

A simple plasmid was first constructed to test the DNA assembly method. Two PCR fragments with complementary overhangs, the DNA encoding for T7 capsid gene *10* and the ColE1 plasmid backbone, were assembled and transformed into *E. coli* cells. The plasmid was successfully recovered without any mutations (Supplementary Fig. 4, Supplementary Data 4). We then re-assembled the T7 WT phage (Fig. 2a) from four PCR amplified fragments (Supplementary Fig. 5) with 50 base pairs (bp) overlaps (Supplementary Data 2). The four DNA parts were mixed at an equimolar concentration in the nanomolar range and annealed according to a published protocol[23]. The assembly reaction was directly added to the TXTL reaction to obtain T7 WT $10^{10}$–$10^{11}$ PFU/ml titers, similar to performing TXTL with a T7 WT genome concentration of 0.1 nM (Supplementary Fig. 6a–c). Importantly, no phages were detected in control experiments expressing any combinations of only three annealed DNA fragments (Supplementary Fig. 6d–f). Unexpectedly, some fragments expressed T7 components that inhibit *E. coli* growth in a dose-dependent manner (Supplementary Fig. 6g, h). This result demonstrates the ability of PHEIGES to explore the activity encoded by the products of genome gene sets.

### Leak-free genomic insertions

Next, we used PHEIGES to insert the *mCherry* reporter gene cassette, consisting of a T7 promoter, a strong RBS, the *mCherry* fluorescent reporter gene and a T7 terminator (Supplementary Data 4) at three different T7 genomic loci (downstream of gene *1*, downstream of gene *10* and of gene *17*; Fig. 2b) to create three T7-mCherry phages. Six gene fragments were amplified using six sets of oligonucleotides designed with overlapping sequences matching the phage T7 and the *mCherry* cassette (Supplementary Data 2). TXTL of the in vitro assembled T7-mCherry genomes yielded a mixture of T7 WT phages and T7-mCherry phages in a ratio of ~1:$10^3$ phages (Supplementary Fig. 7). *mCherry* positive phage strains were recovered by screening individual plaques in 96-well liquid culture format (Supplementary Fig. 8), followed by fluorescence microscopy (Supplementary Movie 1) and NGS sequencing (T7-mC-g10, Supplementary Data 3).

We further improved our workflow to eliminate synthesis of non-edited T7 phages and create a leak-free PHEIGES by adopting a library of orthogonal oligonucleotides[45]. In our first strategy, the overlapping sequences used for annealing the PCR fragments are natural genomic sequences, which may not be strongly orthogonal to start with. Incorporating orthogonal sequences into the oligonucleotides used to generate the PCR fragments decreases crosstalk between overlapping sequences during annealing. Sense and antisense oligonucleotides

## PHEIGES workflow

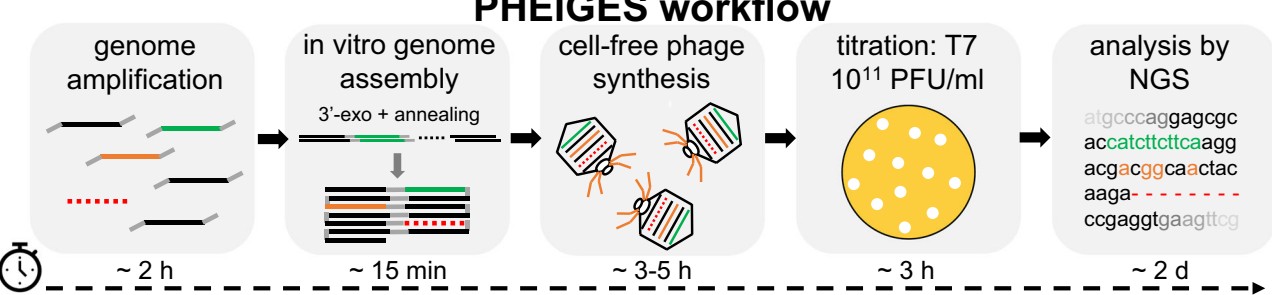

**Fig. 1 | PHEIGES workflow.** The phage genome is amplified by PCR into fragments of 12 kbp or less with overlapping DNA sequences. Gene addition (green), mutation (orange), and deletion (red dotted line) are introduced at any permissive locus. The PCR products are cleaned up, briefly digested with a 3' exonuclease to create sticky ends and annealed in vitro. The DNA assembly reaction is directly added to a CFS reaction to produce phages. Engineered phages are propagated and titrated on the desired host (PFU: plaque forming unit) and directly used for NGS (next generation sequencing) and downstream applications.

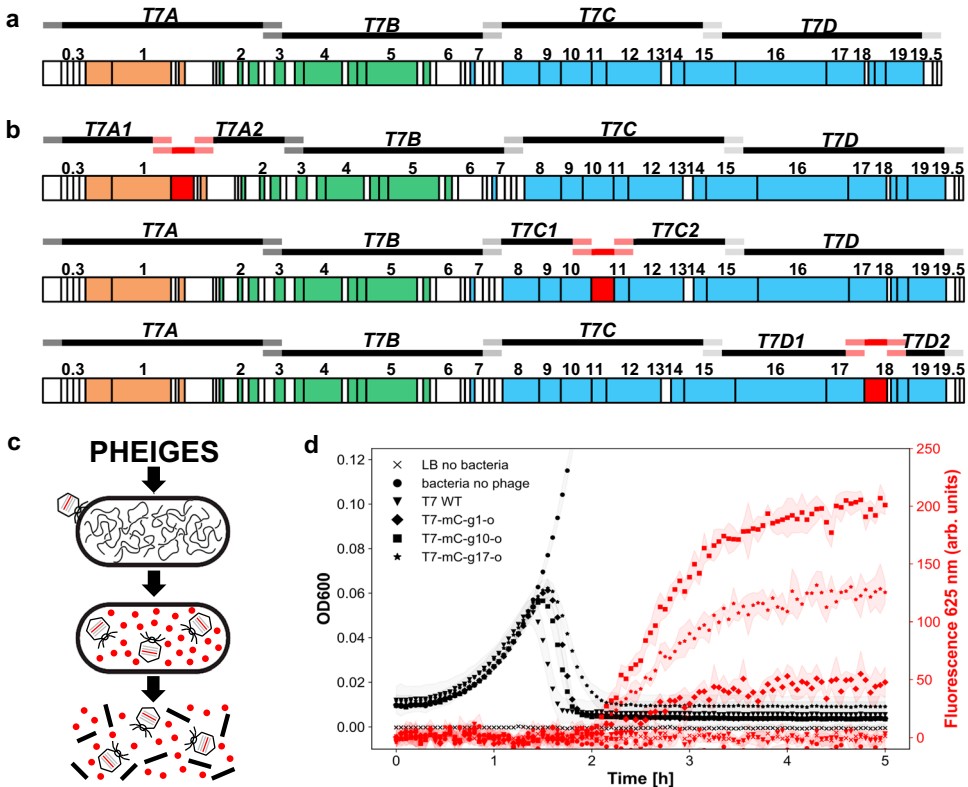

**Fig. 2 | PHEIGES T7 WT genome reconstruction and gene insertion. a** Designated four DNA fragments for T7 WT Genome assembly: *T7A* (1-10700), *T7B* (10700-20500), *T7C* (20500-30600), *T7D* (30600-40000). **b** *mCherry* reporter gene insertion loci downstream of gene: *1* (T7 RNA polymerase), *10* (capsid protein) or gene *17* (tail fiber). **c** T7-mChery life cycle **d** Kinetics of infection of *E. coli* B cultures by T7-mCherry phages. Left: OD600 (dark curves). Right: fluorescence at 625 nm (red curves), arbitrary units. Source data are provided as a Source Data file.

were added to the 5′ and 3′ of the fragments to be annealed. With this design, all the phages detected were harboring the cassette, and no phages were synthesized in the control experiment without the insert (Supplementary Fig. 9), *mCherry* fluorescence signal was detected upon lysis for the three T7-mCherry phages (Fig. 2c, d, Supplementary Fig. 10) and insertion at the three loci was confirmed by NGS (Supplementary Data 3). We employed this primer design approach subsequently and systematically observed no leak in all the control experiments for any type of edits.

## T7 phage genome size reduction

The T7 WT capsid volume <2 kb limit on new gene insertion[16,46] can be relieved by deleting non-essential genome sequences[21,46,47]. Here, we used PHEIGES to reduce the T7 genome length while keeping phage infectibility on its natural host *E. coli* B. We targeted six potential gene segments in the first 20 kbp of the T7 genome to delete non-essential class I genes (early genes) and class II genes (DNA metabolism and replication)[46] (Fig. 3a, Supplementary Fig. 11, Supplementary Data 2). Class III structural genes, considered essential, were not targeted for deletion. The T7 genome was divided into the eight fragments adjacent to the six deletions (del-1 to del-6) (Fig. 3b) that were PCR-amplified with 20 bp-long orthogonal overhangs to bridge the fragments around the deleted regions. With this single design, we could test any combination of deletions. A round of single deletion PHEIGES led to viable phages for del2-4 (but not for del1, 5, and 6; Fig. 3a) that were combined in a second round to create a viable 35.8 kbp T7 ('T7-mini'). In T7-mini, the T7 ligase (in del2) is deleted, indicating that the ligase is not essential for DNA assembly in TXTL. The plaques formed by T7-mini are clear and circular (Fig. 3c). The three deletions were confirmed by whole genome sequencing (Supplementary Data 5). By releasing 4 kbp from the original WT genome of

39.9 kbp, the T7-mini could be re-engineered with insertions of up to 5 kbp.

## Expanding T7 phage host range

Expanding the host range of phages is a major goal of phage engineering[6], serving to unravel their infective mechanism and ecology, and to evolve potent phages for therapeutic and biotechnological applications. Here, we used PHEIGES to create and select variant T7 phages capable of infecting lipopolysaccharide (LPS)-variant ReLPS *E. coli* strains that are not infected by T7 WT.

Rough LPS are the primary and only receptor necessary for T7 WT infection of the natural host *E. coli* B. LPS consists of a lipid A, located in the outer lipid membrane, covalently linked to a polysaccharide chain. The hydrophilic polysaccharide chain consists of a core chain and a repeating O-specific chain. Rough LPS are classified by size from the largest RaLPS to the smallest ReLPS (Supplementary Fig. 12). Different side chain modifications of the LPS can also be found in different *E. coli* strains[48]. Eleven *E. coli* genes are essential for T7 infection, from which nine are implicated in LPS biosynthesis, their deletions giving rise to truncated LPS forms[49]. We verified T7 WT ability to infect these mutants, using a smooth LPS (Seattle 1946, smooth O6[50]) and ClearColi[51] strains devoid of LPS as negative controls. We observed that T7 WT was unable to infect the negative controls, and three of the knockout strains, all harboring the smallest ReLPS (*rfaC*, *lpcA*, and *rfaE*, efficiency of plating (EOP) < 10⁻⁵, Supplementary Figs. 13 and 14). As expected[20], *rfaD* strain had a higher EOP of $4 \times 10^{-4}$. T7 lysate from *E. coli* B, KEIO parent strain BW25113 and TXTL expressed. Similar titers are observed in all conditions. This suggests that *E. coli* B and K12 derivative strains modification-restriction systems do not affect T7 EOP and that T7 interacts with *E. coli* B and *E. coli* BW25113 in the same way. Variable T7 WT EOP was recorded on *E. coli* B (as T7 WT shows

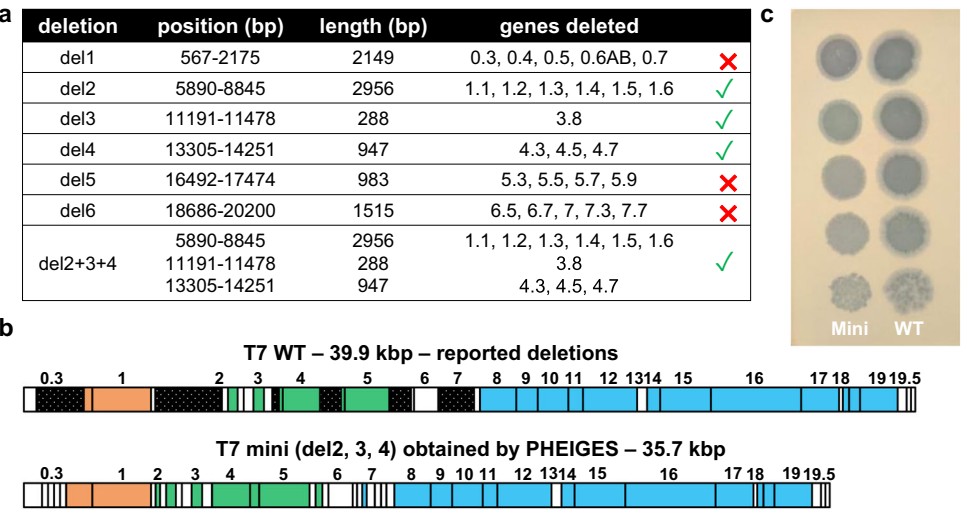

**Fig. 3 | T7 WT (wild type) genome size reduction using PHEIGES. a** Assayed six deletions and combinations thereof (bp: base pair). **b** Schematic mapping of attempted deletions (black) on T7 WT and of of T7-mini (kbp: kilo base pair). **c** Image of the plate showing the spots of a 10-fold serial dilution of $10^8$ PFU/mL T7 WT (left) and T7 mini (right) lysates on an *E. coli* B lawn after overnight of incubation at 37 °C.

identical EOP on Keio parent strain *BW25113*) and the other knockout strains as reported in Supplementary Fig. 14.

Previous work identified a T7 mutant phage capable of infecting ReLPS *E. coli* while retaining its original LPS phenotype with mutations in the tail genes *11* and *12* and in the tail fiber gene *17* [52]. Concordantly, we selected a plaque from a natural T7 WT phage variant infecting *rfaD E. coli* strain (T7-rfaD-1) harboring two mutations (G784E, in the tail gene *12*; S541R, in the tail fiber gene *17*, Supplementary Data 1, 3). T7-rfaD-1 was able to infect the other LPS knockouts strains (*rfaC*, *lpcA*, and *rfaE*) as well as the natural host *E. coli* B strain. We verified that the T7-rfaD-1 variant interacts with ReLPS with an in vitro assay. We incubated $10^9$ PFU/mL lysates of T7 WT or T7-rfaD-1 phages with either purified water, RaLPS, smooth LPS, or ReLPS lipopolysaccharide variants at 0.2 mg/ml (Supplementary Fig. 15). The Phage-LPS mixtures were then titrated on *E. coli* B to reveal the remaining phages. As expected, T7 WT titers remained unchanged following pre-incubation with water, smooth LPS, and ReLPS. Concomitantly, no phage was detected following preincubation with RaLPS indicating that all the T7 WT phages ejected their genomes upon reaction with RaLPS [53]. In contrast, no phage was detected following pre-incubation of T7-rfaD-1 with purified ReLPS or RaLPS (Supplementary Fig. 15). We estimated that the T7-rfaD-1 variant is at least ~10 000 times more sensitive to ReLPS in vitro than T7 WT. This confirms that T7-rfaD-1 infects *rfaD*, *rfaC*, *rfaE* and *lpcA* strains through interaction with ReLPS receptors. This assay provides credence for quantifying phage ReLPS infectious phenotypes in vitro.

### Evidence of a phenotype/genotype linkage in batch TXTL

The ability of phages to couple their genome to their encapsulating proteins gave rise to a plethora of applications as phage display [2,47]. Under natural conditions, coupling is due to the limited number of phages (typically one) that infect and propagate in a single bacterial host cell, assuring that phage-encoded proteins are assembled with their own coding genome. In laboratory settings, genetic phage variant libraries are either directly transformed into host cells, or they are first packaged in vitro and then used to infect host cells to reveal their linked phenotypes. The only current methodology dispensed of a compartmentalization is Ribosome Display. The latter is limited by the fragility of the mRNA-ribosome-polypeptide linkage and the fragility of mRNA [54]. PHEIGES achieves phage synthesis without compartmentalization. We reckoned that the fast kinetics of T7 phage coat and tail

proteins' cooperative assembly to encompass the phage genome, following their coupled transcription and translation, may limit their diffusion and cross-binding to non-self-phage genome in the viscous TXTL mix, leading to genotype to phenotype linkage without the need for encapsulation or in vivo infection and propagation.

To determine whether phage genotype/phenotype (g/P) coupling prevails in PHEIGES in absence of physical compartmentalization, we devised a simple two-phage experiment consisting of co-expression of two phages whose genomes are annealed in the same DNA assembly reaction. Both phages, T7-vWT (T7 variant WT phage infecting only *E. coli* B, WT host) and T7-rfaD-1 phage, carry gene *12* G784E. T7-rfaD-1 carries an additional mutation (gene *17* S541R) rendering it infectious on both *WT* and ReLPS *rfaC* hosts (Supplementary Data 1, 3). Co-assembly was done with an equimolar mix of *T7D* and *T7D-S541R* fragments (Fig. 4a). Expression of this one-pot dual genome assembly mix in TXTL reactions is expected to result in six different g/P types of phages in absence of linkage, where "MIX" represents a mixture of incorporated WT and S541R mutant tail fibers (Fig. 4a).

We calculated the theoretical fully randomized relative abundance of the six different g/P hybrids under the most permissive assumption (H0, Supplementary Note 1, Supplementary Fig. 16), namely that the integration of at least one S541R mutant monomer into one of the six trimeric tail fibers of a given phage is necessary and sufficient to infect *rfaC* ReLPS (Fig. 4d). The following set of experiments allowed us to quantify the abundance of the six phages g/P hybrids in the co-synthesis of equimolar assemblies of T7-vWT and T7-rfaD-1 (Fig. 4d, Supplementary Fig. 17).

The fraction of mut/MUT and mut/MIX phages (Fig. 4b), 50 ± 18%, was derived by titrating the equimolar co-expression on *WT* and *rfaC* strains ($4 \pm 1.0 \times 10^8$ and $8 \pm 1.0 \times 10^8$ PFU/mL, respectively), as wt/WT or mut/WT cannot infect the *rfaC* strain while wt/MIX or wt/MUT cannot propagate on strain *rfaC* after infection. As controls, co-synthesis of separately assembled T7-vWT and T7-rfaD-1 phages resulted in the expected 50% of *rfaC* vs *WT* infection ($2.8 \pm 0.9 \times 10^8$ and $6 \pm 1.4 \times 10^8$ PFU/mL, respectively). While the EOP ratio of separately synthesized T7-rfaD-1 between *rfaC* and *WT* hosts ($5.8 \pm 0.7 \times 10^7$ and $3.2 \pm 0.6 \times 10^8$ PFU/mL, respectively) was 0.18 ± 0.04, no plaques from separately synthesized T7-vWT were detected on the *rfaC* strain indicating that <$10^4$ PFU/mL of ReLPS emerging variant phages were present in the T7-vWT assembly. As expected, we found comparable phage titers on *WT* host for T7-rfaD-1, T7-vWT or their equimolar

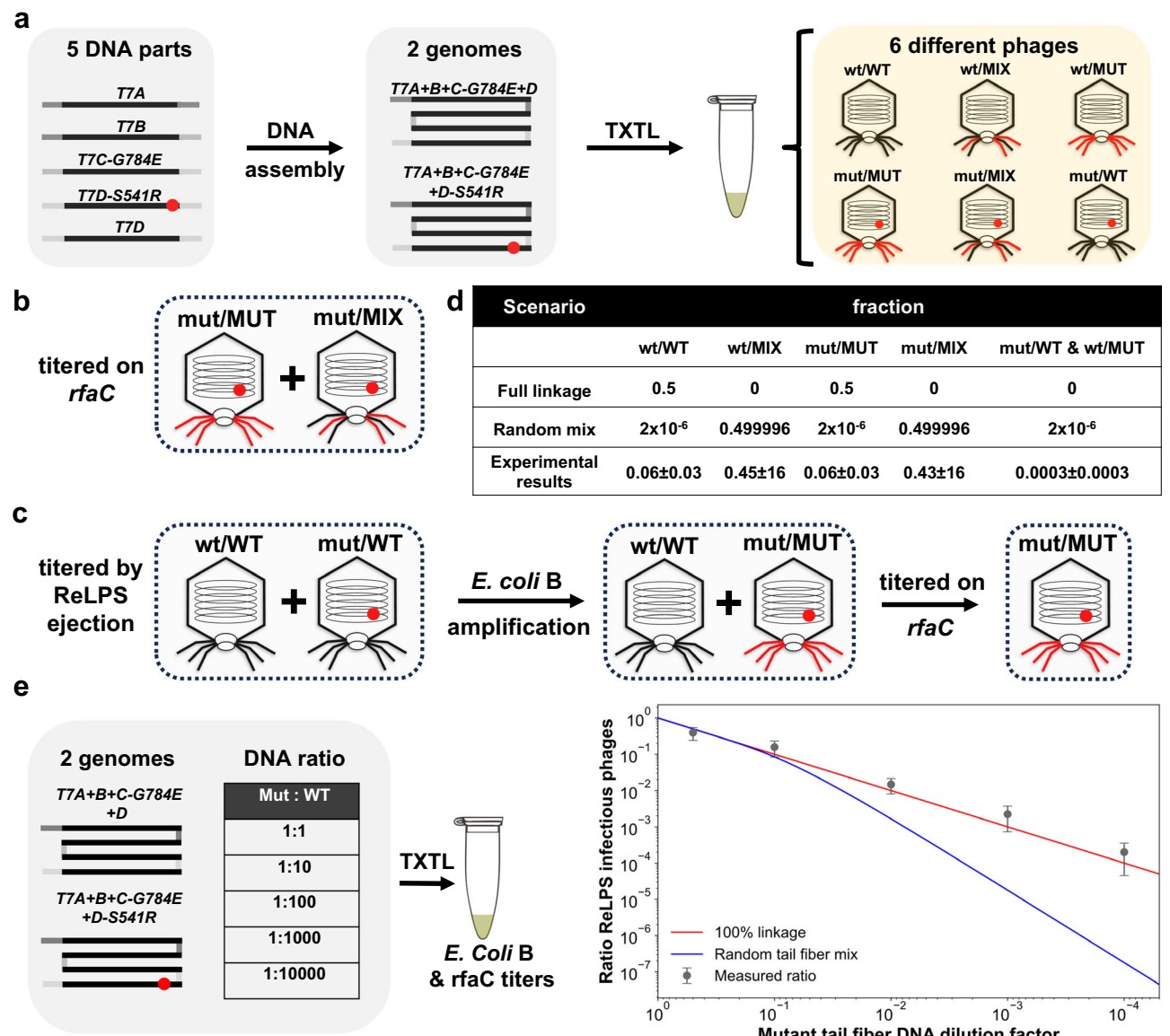

**Fig. 4 | Evidence of a g/P linkage in batch TXTL (cell-free transcription-translation) reactions. a** One pot assembly of T7 phages from five parts leading to two possible phage genomes carrying a tail fiber mutation or not. Batch TXTL expression of this equimolar co-assembly leads to six types of phages, phages encapsulating a mutant tail fiber genome or not and displaying combinations of mutant and wild-type tail fibers (mut: mutant genotype, MUT: mutant phenotype, wt: wild type genotype, WT: wild type phenotype, MIX: mixture of MUT and WT) **b** Titer of the co-expression on *rfaC* versus *E. coli* B determines the mut/MUT + mut/MIX proportion of phages in the co-expression. **c** Presence of purified ReLPS leads to selective inhibition of phages with mutant tail fibers. *E. coli B* titer of the phage/LPS mixture determines the wt/WT + mut/WT initial proportion of phage in the co-expression. Subsequent amplification on *E. coli B* transforms mut/WT phages in mut/MUT phages. Titer on *rfaC* and *E. coli* B determines the mut/WT over wt/WT proportion of phages in the ejection mixtures. **d** Table compiling the proportion of phage types measured in the co-expression and the predicted proportion for a random tail fiber mix with the hypothesis that the six trimer tail fibers of a phage results from the random combinations of three monomers. **e** Co-expression of mutant and WT tail-fiber phages with various decreasing proportion of mutant tail fiber. The random tail mix corresponds to the H0 hypothesis while other hypotheses regarding the random assembly of tail fibers are also considered (Supplementary Fig. 16). *n* = 3 biologically independent experiments. Data are presented as mean values +/− SD. Source data are provided as a Source Data file.

co-synthesis or their equal volume of their separate synthesis (~6 × 10⁸, 3 × 10⁸, 8 × 10⁸, 6 × 10⁸ PFU/mL, respectively).

The fraction of wt/WT and mut/WT (Figs. 4c), 6 ± 2%, was determined by amplifying and titrating the *WT* 50 ± 18%, infective phages, following incubation of the equimolar co-expression phage mix with purified ReLPS (0.4 mg/mL, 37 °C overnight), in comparison to an equal volume mix of the separately assembled genes (1.9 ± 0.4 × 10⁶ for equimolar, 1.7 ± 0.4 × 10⁷ PFU/mL for equivolume). Under such conditions, all other variants are phenotypically ReLPS+ and eject their genome. Indeed, no plaques on *rfaC* were detected in the ReLPS overnight incubation of the T7-rfaD1 mutant tail fiber expression, the

co-expression, and the equal volume mix. Separately assembled T7-vWT phages were poorly susceptible to ReLPS when incubated with 0.4 mg/mL ReLPS overnight at 37 °C (10-fold decrease of titer from ~6 × 10⁸ PFU/mL to ~5 × 10⁷ PFU/mL).

Finally, the fractions of wt/WT (Fig. 4c), 6 ± 2%, and mut/WT, 0.03 ± 0.02%, were determined by amplifying on *WT* host and titrating the wt/WT and mut/WT fraction (above) on *WT* and *rfaC* hosts (6 ± 2 × 10¹⁰ and 2 ± 1 × 10⁷ PFU/ml, respectively). A few undefined plaques (negligible percentage compared to co-expression) were also detected on *rfaC* from the equal volume mix of pre-assembled T7-vWT and -rfaD-1 phages after the ReLPS ejection and amplification on *WT*

host. We attribute these plaques to new mutant phages emerging from T7-vWT.

The above results together with the following two hypotheses based on the symmetry due to equimolarity and considering that the point mutation does not entail tail fiber assembly bias:

$$
\begin{aligned}
&(wt/WT)/(wt/WT + wt/MIX + wt/MUT) \\
&= (mut/MUT)/(mut/MUT + mut/MIX + mut/WT)
\end{aligned} \quad (1)
$$

$$
\begin{aligned}
&(wt/MUT)/(wt/WT + wt/MIX + wt/MUT) \\
&= (mut/WT)/(mut/MUT + mut/MIX + mut/WT)
\end{aligned} \quad (2)
$$

Equations (1) and (2) describe the assumptions that the ratio of pure p/G phages (mut/MUT and wt/WT) as well as opposing p/G (mut/WT and wt/MUT) on all the total number of phages that carry a mut or a wt genotype are the same. We define pure p/G as a phage that displays only tail fibers that correspond to its genomic sequence, for example, wt genotype and only WT tail fibers. Conversely, we define as opposing p/G a phage that displays only tail fibers that do not correspond to the genomic sequence for example wt genotype and only MUT tail fibers. These minimal assumptions allowed us to determine the relative fractions of the six phage variants expected from the initial co-assembly mix (Fig. 4d). A clear disparity is evident between the most permissive randomized hypothesis and the experimental results, indicating significant g/P coupling. The fraction of pure g/P (wt/WT and mut/MUT) coupling amounts to 12% of what is expected from full linkage and four orders of magnitude greater than the randomized hypothesis. g/P coupling is also manifested by the stark (x200) asymmetry between pure and opposing p/Gs.

Next, we tested whether we could selectively enrich T7-rfaD-1 phages from PHEIGES assembly as above but with different ratios (1:2 – 1:10,000) of *T7D-S541R* to *T7D-WT* at constant DNA concentration (50 pM). The ratio of plaque counts of the resulting phages between *rfaC* (T7-rfaD-1 phages), and *WT* (both phages) hosts closely follows the dilution regime, as expected from g/P linkage, and not the hypothetical random assembly regime (Fig. 4e, Supplementary Fig. 18). Taken together, these experiments support the existence of local interaction between the tail fibers, the procapsid and their encoding genome, indicative of localized encapsulation underlying the observed g/P coupling. Calculations and other hypotheses are compared in Supplementary Note 1 and Supplementary Fig. 16.

**Selection for T7 phage host range expansion via PHEIGES g/P linkage**

The tip of the tail fiber gene *17* (amino acids 472–554), determinant of the phage's host range[49,55,56], can be exchanged between phages or mutated to adapt to new hosts[17,57–61]. Recently[20], a library of 1660 exhaustive single mutations of the T7 tail fiber tip residues was probed using yeast cloning, site-specific recombination with a helper plasmid, Cas9-gRNA-based progressive variant phages, and final amplification of the phage variant library in absence of helper plasmid. Tail fiber mutations rendering T7 variants infectious to truncated LPS *E. coli* strains *rfaD* (ReLPS) and *rfaG* (Rd1LPS) were identified (Supplementary Data 5). In another study, a continuous evolution setup generated T7 variants with tail and tail fiber gene mutations specific to different rough LPS types[52] (Supplementary Data 5). Here, we retrieved similar mutation patterns with the simplified, faster, and cheaper PHEIGES protocol (Fig. 5a, b).

A fragment consisting of the 172 C-terminal residues (516 bp) of the T7 WT tail fiber was amplified by high-fidelity PCR (fragment E0). We used fragment E0 to create 3 libraries of randomly mutated fragments with increasing PCR mutational load (E1, E2, and E3; see Methods). NGS analysis ($2 \times 10^5$ reads/fragment) of the raw sequences

revealed a uniform distribution of mutations averaging 1, 2.5, and 5 mutations per fragment for E1, E2, and E3, respectively (Fig. 5c). PHEIGES was used to assemble these libraries with the rest of the genome, amplified by PCR in five overlapping parts, resulting in four T7 variant phage pools: unmutated T7-E0, and T7-E1, T7-E2, and T7-E3 libraries, with diminishing respective titers on *WT* host of $1 \times 10^{11}$, $3 \times 10^{10}$, $1 \times 10^{10}$ and $7 \times 10^9$ PFU/ml (as compared to $1 \times 10^{11}$ of rebooted T7 WT), suggesting an overall detrimental mutational load (Supplementary Fig. 19). No phages were detected in a control in absence of the tail fiber fragment in the assembly mix. Given the established PHEIGES g/P linkage, the three libraries were directly spotted on our panel of strains (Supplementary Figs. 20 and 21). As expected, none of the phage libraries infected the LPS-deficient smooth or ClearColi *E. coli* strains. Hundreds of plaques were obtained from T7-E0, -E1 and -E2 libraries infecting previously described LPS phenotypes of rfaG (Rd1LPS) and rfaD[20] (Supplementary Fig. 21).

Importantly, we successfully and systematically obtained ~100 clear and circular plaques per 1 ng of T7-E1 and T7-E2 TXTL-assembled libraries by direct plating on ReLPS *rfaC*, *lpcA* and *rfaE E. coli B* hosts that were not infected by T7 WT (or the T7-E0 control). This is in stark contrast to our estimation of the probability of selecting a single infective library genetic mutant that might have randomly packaged with an infecting phenotype in a well-mixed milieu and assuming hundred possible infective mutations of an estimated $2 \times 10^6$ variants per 1 ng assembled library ($< 6 \times 10^{-3}$ PFU). This strongly supports a g/P linkage in PHEIGES.

Twelve clonal phages from each of the three hosts (Fig. 5d–f) were purified, propagated in their respective host, phenotyped on all hosts (Supplementary Figs. 22–24) and sequenced (Supplementary Fig. 25). Unlike T7 WT, all mutants exhibited a broad infection spectrum. Interestingly, some T7 variants isolated from ReLPS strains could not infect the *rfaF* host. This observation is reminiscent of a contraction of the host range discussed previously[52].

Sequencing of the 36 clones (500 bp upstream and downstream of the inserted tail fiber fragment) revealed more amino acid mutations in T7-E2 and T7-E3 as compared to T7-E1. The mutations were located mainly in the exterior loops of the tip of the tail fiber (Fig. 5d–f), matching the top five residues (G521, S541, A500, N501, G480, D540) discovered previously by systematic mutagenesis for *rfaD* infection[20]. Moreover, we identified four common substitutions: G521R, N501K, S541R, G480R. Comparable results are found for *rfaG* (Supplementary Data 5). This suggests that PHEIGES workflow can quickly generate and identify the best gain of function variants. The presence of silent mutations within the tail fiber gene assures that these clones resulted from the error prone libraries and did not appear de novo when propagating on their host. Whole genome sequencing of six of the phage clones (two for each of *rfaC*, *lpcA*, and *rfaE* mutant *E. coli* strains) identified further mutations in tail genes *11* and *12*[49,52] (Supplementary Data 3). PHEIGES allowed us to reconstruct separately gene *11* and gene *12* variants and gene *17* variants, to demonstrate that ReLPS strain infection was solely dependent on the selected gene *17* mutations (Supplementary Figs. 26, 27, Supplementary Data 3). Subsequent mutations in the tail genes *11* and *12* provide increase the fitness of the phages within the new host.

PHEIGES presents several advantages compared with ORACLE[20] and other current methods[16,22,23,62] in generating and identifying phages with improved host ranges. Firstly, the library approach to generate infectious phages can be performed on any large genetic parts of the phage within a day as compared to ORACLE that necessitates additional cloning and downstream selection steps. Secondly, PHEIGES has the capability to address multiple amino acid mutation per sequence by manipulating the mutagenic PCR parameters. Thirdly, PHEIGES provides greater versatility through its all-cell-free method. In ORACLE, tail fiber residues are classified based on the relative abundance each variant when the library is propagated into

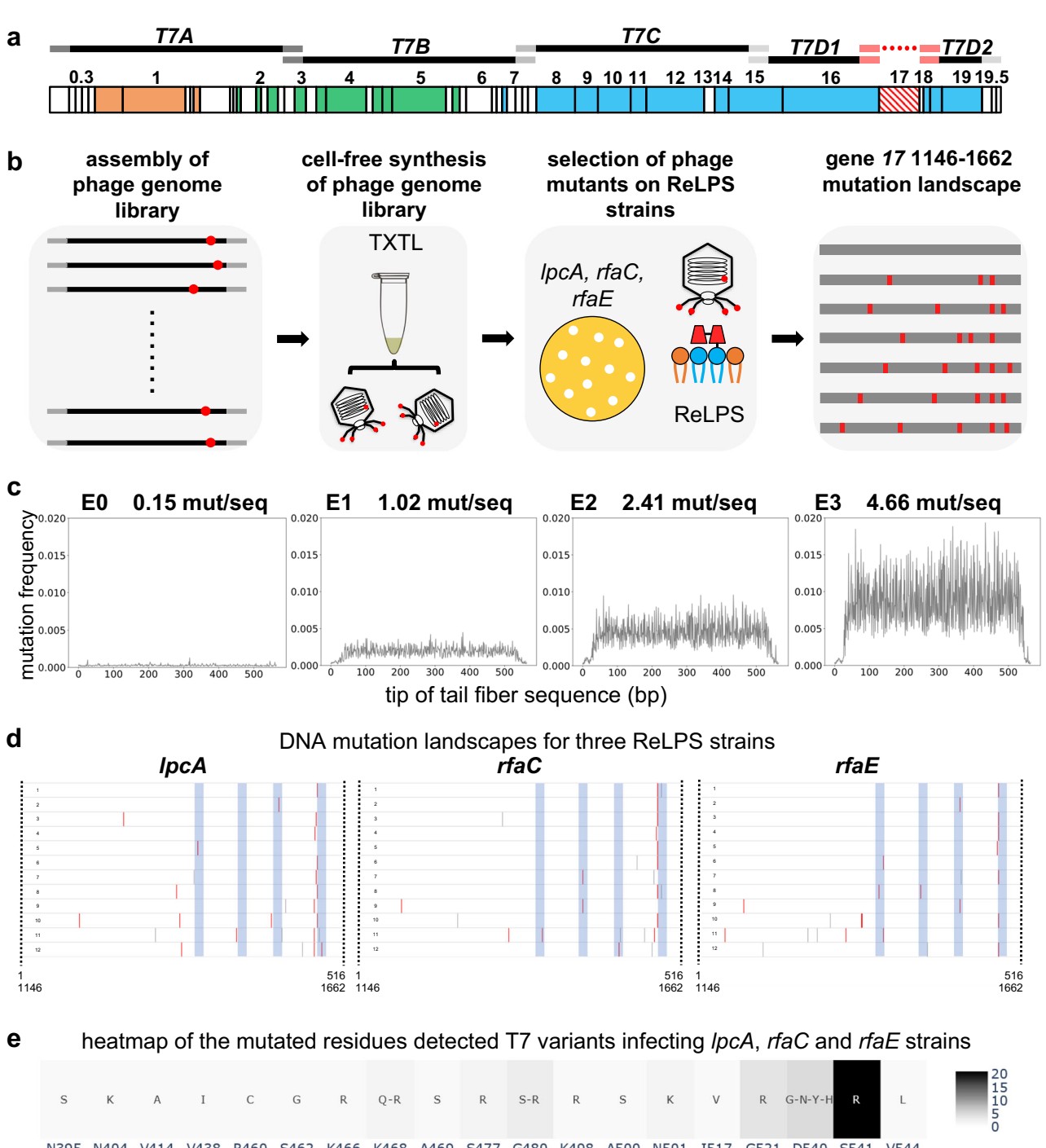

**b** | assembly of phage genome library | cell-free synthesis of phage genome library | selection of phage mutants on ReLPS strains | gene *17* 1146-1662 mutation landscape

**c** E0 0.15 mut/seq · E1 1.02 mut/seq · E2 2.41 mut/seq · E3 4.66 mut/seq — mutation frequency vs tip of tail fiber sequence (bp)

**d** DNA mutation landscapes for three ReLPS strains — *lpcA*, *rfaC*, *rfaE*

**e** heatmap of the mutated residues detected T7 variants infecting *lpcA*, *rfaC* and *rfaE* strains

S N395 · K N404 · A V414 · I V438 · C R460 · G S462 · R K466 · Q-R K468 · S A469 · R S477 · S-R G480 · R K498 · S A500 · K N501 · V I517 · R G521 · G-N-Y-H D540 · R S541 · L V544

**f** most frequent mutations found in T7 variants infecting *lpcA*, *rfaC* and *rfaE* strains

| Mutation | S541R | G521R | D540N/Y/H/G | S477R | G480R/S | N501K |
|---|---|---|---|---|---|---|
| Counts | 21/36 | 4/36 | 5/36 | 2/36 | 3/36 | 2/36 |
| Strains | *lpcA rfaC rfaE* | *lpcA rfaC rfaE* | *lpcA rfaC* | *lpcA rfaE* | *rfaC rfaE* | *rfaC* |

the host. While probably correct as a first approach, other naturally occurring mutations in the tail fiber as well as the tail complex are omitted. The phage variants in higher abundance detected in ORA-CLE were likely to also carry mutations, for instance, in the tail genes as reported here (Supplementary Data 5) and in other works[49,52]. Analyzing these other mutations for each variant (1660 variants) would have been too cumbersome and sequence extensive. With PHEIGES, we show that while other mutations appear in the tail

genes, the tail fiber mutations are the major determinants of the host range. By re-assembling phages from the selected variants with improved host range with tail fiber-only mutations and phages with tail-only mutations, we show that the tail fiber-only mutations are sufficient to infect *ReLPS* strains, whereas mutations in the tail genes are not. Fourthly, PHEIGES is compatible with phage library assembly with or without cellular amplification. For the exploration of gain of function variants, cellular amplification is dispensable given the high

**Fig. 5 | Using PHEIGES to engineer T7 phages that infect E. coli strains with any type of rough LPS. a** The T7 genome is assembled using five T7 WT gene parts (*T7A, T7B, T7C, T7D1, T7D2*) and the fragment 1146–1662 of the tail fiber gene *17* obtained by mutagenic PCR. **b** PHEIGES assembly workflow for T7 LPS mutant libraries. T7 genomes with E0, E1, E2 and E3 were separately assembled and expressed in TXTL (cell-free transcription-translation). TXTL reaction containing the phage libraries were directly spotted on *E. coli* strain harboring ReLPS. Twelve T7 variants were selected, and their tail fiber sequenced for each of the three *E. coli* mutant strains *lpcA*, *rfaC* and *rfaE* (all ReLPS), 36 phages total. *E. coli* B strain has a type RbLPS. **c** The mutation frequency of the four mutated tail fiber DNA fragments was determined by NGS. The graphs are indexed from 1 to 516 which corresponds to the bp 1146–1662 of gene *17*. Source data are provided as a Source Data file. **d** The

tail fibers of the selected phages were sequenced to establish the mutation landscape, especially the T7-ReLPS. The blue zones show the external loops of the tail fiber tip. Gray bars are silent mutations, red bars are non-silent mutations. 6 plaques were picked from T7-E1 (6 first from the top in Fig. 5d, labeled 1–6), 3 plaques from T7-E2 (three following in Fig. 5d, labeled 7–9), and 3 plaques from T7-E3 (last three in Fig. 5d, labeled 10–12) from *rfaC*, *lpcA*, and *rfaE*. Source data are provided as a Source Data file. **e** This heatmap compiles all the mutations of the 3 × 12 phages. the native amino acid is indicated below every mutation and the color scale indicates how many times a mutation is found in the 36 variants. **f** This table reports the most frequent amino acid mutations among the 36 T7 variants that lead to ReLPS infection as a single mutation.

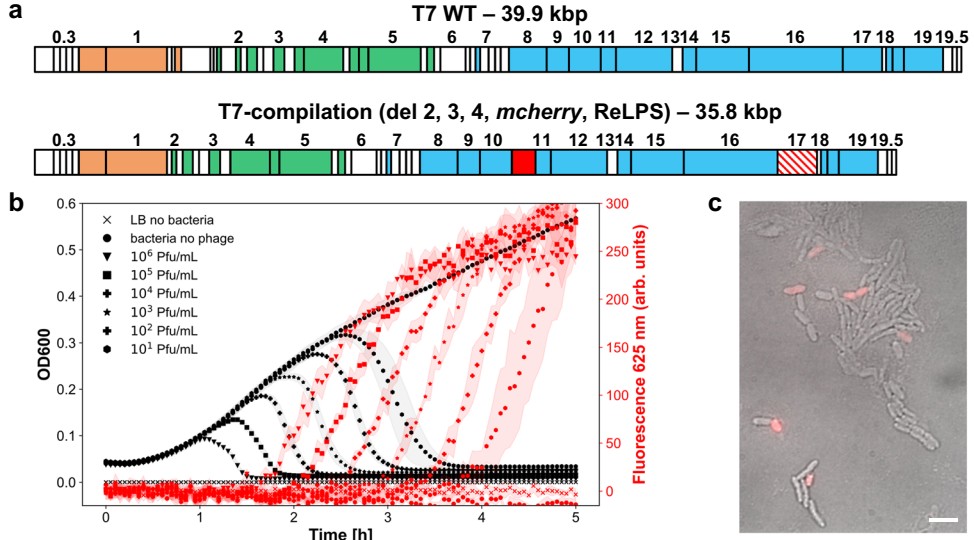

**Fig. 6 | Using PHEIGES to engineer a T7-compilation genome. a** The T7-compilation genome incorporates three edits: deletions (del 2, 3, 4), insertion of the *mcherry* gene after gene *10* and mutation of the tail fiber to infect ReLPS strains (WT: wild type). **b** Kinetics of infection of *E. coli* mutant strain *rfaC* cultures by the T7-compilation phage. Left: OD600 (dark curves). Right: fluorescence at 625 nm (red curves). Source data are provided as a Source Data file. **c** Microscopy image of *E. coli* mutant strain *rfaC* infected with the T7-compilation phage (merge of phage contracts and fluorescence channels). Scale bar corresponds to 5 µm. This experiment was repeated twice with similar results.

phage titers attained by PHEIGES (>10$^{10}$ PFU/mL). Alternatively, if a total p/G is desired, cellular amplification of the TXTL phage library remains an option. Fifthly, PHEIGES addresses the common challenge encountered in editing methods like CRISPR, where there is often an overwhelming prevalence of wild-type phage compared to variants. With PHEIGES, the absence of fragments results in the absence of phages. Consequently, a library of fragments leads to a library of phages with the relative abundance of each phage variant directly proportional to the relative concentration of DNA fragments. Finally, we anticipate that PHEIGES could enable radical phage hybridization and genome reshuffling, overcoming limitations seen in conventional co-infection.

### Peptide display using PHEIGES
In order to pursue in the future in vitro selection based on PHEIGES, we tested whether PHEIGES could successfully assemble phages with peptide fusions to either the main coat protein (gp10b) as is commonly practiced[47] or to the tail fiber protein (gp17). To this end, we used as a model a triple repeat of the FLAG tag peptide (DYDDDDK). Our results (Supplementary Fig. 28) demonstrate successful phage assembly of both gp10b and gp17 FLAG fusions. Notably, T7 phages displaying gp17-FLAGx3 fusion, retained their infectibility, suggesting the fusion does not hinder the interaction between the tail fiber and the LPS host receptor.

### PHEIGES compilation
To demonstrate PHEIGES rapid, iterative use, we assembled in one step T7-compilation with minimal genome (T7 mini), *mCherry* insertion downstream of gene *10* and tail fiber ReLPS infective mutation (Fig. 6a, Supplementary Fig. 28) using the corresponding PCR fragments. The resulting phages were confirmed by NGS sequencing (Supplementary Data 5) and phenotyping (Fig. 6b, c, Supplementary Fig. 29). Lysis kinetics of T7-compilation were slower as compared to the ReLPS infective phage indicative of an associated fitness cost (Supplementary Fig. 30).

### Towards non-model phages
This work is based on the model T7 phages to establish PHEIGES workflow. T4 phage (~169 kbp) was previously expressed from its purified genomes in *E. coli* TXTL[38]. Subsequently, we reproduced T4 TXTL expression (Supplementary Fig. 32) and, for the first time, demonstrated the expression of *E. coli* phages VpaE1 (~88 kbp) T6 (~170 kbp) along with *Salmonella* phages FelixO1 (~87 kbp) and S16 (~169 kbp) from their purified genomes at 0.1 nM in TXTL. These results suggest the expression capability for a spectrum of *E. coli* and non-*E. coli* phages with genomes ranging from 40 kbp to 170 kbp in *E. coli*-based TXTL. To assess scalability, we applied PHEIGES to *Salmonella* phage FelixO1. We re-assembled FelixO1 genome from five PCR fragments directly amplified from a salmonella phage lysate.

We obtained ~$10^8$ PFU/mL of FelixO1 phages (Supplementary Fig. 33, Supplementary Data 1, 2). This result suggests that PHEIGES could be applied to engineer a range of non-*E. coli* phages with larger genomes than T7. To our knowledge, FelixO1 is the largest phage genome rebooted in vitro. Ongoing investigations will elucidate the applicability of PHEIGES for engineering larger phages with chemically modified genomes like T4 and T6 phages or non-*E. coli* phages that may require alternative expression platforms beyond *E. coli*-based TXTL.

## Discussion

PHEIGES provides a rapid, technically accessible, and low-cost method for phage engineering. Its DNA assembly efficiency (~$10^{10}$ PFU/ml) offers many advantages compared to the current in vivo methods[15,18,19] and other cell-free phage engineering methods[21,40,63]. The yeast cloning approach to phage genome engineering takes on the order of one week to achieve[64,65]. CRISPR approaches take at least one week to achieve as plasmids must be prepared for homologous recombination first[13,14]. The Golden Gate Assembly method (GGA) takes about two to three days to carry out without taking into account the step of whole genome synthesis to remove unwanted GGA sites[63]. Taking into account the genome synthesis, the GGA method takes more than one week. The GGA method is also not compatible with TXTL, consequently the assembly reaction must be cleaned up first, a step that considerably reduces the yields[63], which is also the major limitation to the Gibson assembly method for phage engineering (<$10^5$ PFU/ml)[40]. The PHEIGES workflow presented in this work eliminates all these steps. PHEIGES facilitates multiplexing phage genome engineering by enabling concurrent gene deletions and insertions at any permissive loci into the T7 genome, serving as proof of concept for the versatility and accessibility of this workflow. We anticipate that PHEIGES could be applied to any phages proven to be cell-free synthesized such as non-model and obligate lytic phages. We show that in addition to T7 and T4[38], the coliphages T6 and VPAE1[66], as well as the Salmonella phages FelixO1[67] and S16 are cell-free synthesized with our TXTL system from their purified genomes (Supplementary Fig. 31).

PHEIGES enables synthesizing and selecting T7 phages with engineered tail fiber because a significant proportion of their phenotypes are linked to their genotypes in batch mode TXTL reactions (p/G linkage). Without such a linkage (e.g., capsid protein and other phages yet to be tested), one could explore a library of PHEIGES-engineered phages by selecting through a single passage in vivo. This method, however, has potential biases. A single pass in vivo, for instance, would kinetically bias the selected phages to the infection process (e.g., binding kinetics, binding strength). Without an in vivo step, if the library is not p/G linked all the solutions are represented. In that case, the issue is how to cope with the false positives. Future works will focus on studying the nature of p/G linkage in TXTL and how to improve it. Regardless, PHEIGES is compatible with in vivo passages, either after DNA assembly by transformation or after cell-free synthesis, that could or could not be used depending on the downstream application. If the goal is to produce non-infectious phages for biotechnology applications, such as vaccine scaffolds[12], it is not necessary to pass in vivo. If the goal is to produce infectious phages, passing in vivo can be done but does not necessarily present advantages. PHEIGES already maximizes the exploration of mutations with a mutation rate that can be tuned at will during the PCR amplification. Phages are selected at the last step, thus minimizing in vivo biases. Because the cell-free synthesis is not compartmentalized, PHEIGES produces a myriad of chimeric phages at the structure level. These steps are efficient and show minimal constraints compared to the propagation in living cells (e.g., difficult-to-perform phage genome transformation followed by living cell amplifications with various anti-phage systems and metabolisms). Lastly, PHEIGES enables the synthesis of some non-coli phages. FelixO1, for instance, is a salmonella phage requiring a BLS2 laboratory.

PHEIGES enables engineering and synthesizing phages in BSL1 conditions accessible by all laboratories. As such, PHEIGES extends accessibility to phage engineering.

The g/P linkage may stem from either (i) an evolutionary T7 trait minimizing protein production to the benefit of maximizing phage production from limited cellular resources. A secondary selective advantage may be to accomplish g/P coupling within a cell, where phage mutants may arise due to replication errors; (ii) diffusion constraints of fast oligomerizing system in viscous TXTL with volume per phage (~$10^{-13}$ L) two orders of magnitude larger than the bacterial cell volume (~$10^{-15}$ L). Further studies to differentiate between these hypotheses will determine the ability to generalize the g/P linkage to other phages. Similarly, future work will inform on T7 assembly mechanism and will address the extent to which PHEIGES could be harnessed and generalized for phage display and other applications. Eliminating the need for bacterial compartmentalization opens many possibilities to explore such as further genome size reduction, larger exogenous gene integration, display of biomedically-relevant peptides and unnatural amino acid integration, among other potential benefits.

## Methods
### Reagents
The genomic DNA of phage T7 was purchased from Boca Scientific (# 310025). The DNA ladder for DNA electrophoresis was purchased from Invitrogen (10-787-018). The exonuclease III was purchased from NEB (#M0206S). DNA oligonucleotides were ordered from Integrated DNA Technologies (IDT), standard desalting. The oligonucleotides sequences are in Supplementary Data 2. The bacterial strains were obtained from various sources as described in Supplementary Fig. 12. Plasmid DNAs were obtained from various sources as described in Supplementary Data 4. The LPS EH100 Ra (L9641) mutant and smooth LPS from *E. coli* 0111:B4 (L5293) were purchased from Sigma-Aldrich. The ReLPS was purchased from Avanti Polar Lipids (Kdo2-Lipid A (KLA), 699500). The phage lysates were sterilized with 0.22 μm centrifuge filter tubes (Costar #8160). The phage serial dilutions were performed with filter tips (Dutscher, #014210, #014220). The phage kinetics were performed in flat bottom transparent 96 bacterial culture well-plates with lids (Thermo Scientific Optical-Bottom Plates, #265301) on a Synergy H1 multi-mode microplate reader (Agilent). Phage spotting was performed on square Petri dishes (Greiner Bio-One #688102). Microscopy was done with an Olympus IX81 inverted epifluorescence microscope mounted with a thermoplate (Tokai Hit). An imaging spacer (Grace Bio-Labs #654006) and microscopic slides (Fisher, #12-550-A3), and cover slides (Fisher #12542C) were used for bacterial lysis microscopic experiments. Purchased T7 genome (39.9 kbp, GenBank V01146.1 [https://www.ncbi.nlm.nih.gov/nuccore/V01146], Supplementary Fig. 4) was verified by NGS (Supplementary Data 1) observing two mutations compared to GenBank V01146.1 [https://www.ncbi.nlm.nih.gov/nuccore/V01146]: (i) an insertion of an A base after position 1896 in the gene *0.7*; (ii) A to G mutation at position 22629 (N227S) in gene *9*.

### Accession numbers
T7: GenBank V01146.1 [https://www.ncbi.nlm.nih.gov/nuccore/V01146]. T4: GenBank: NC_000866 [https://www.ncbi.nlm.nih.gov/nuccore/NC_000866.4]. T6: GenBank: NC_054907 [https://www.ncbi.nlm.nih.gov/nuccore/NC_054907.1]. VpaE1: GenBank: NC_027337.1 [https://www.ncbi.nlm.nih.gov/nuccore/849250042]. FelixO1: [https://www.ncbi.nlm.nih.gov/nuccore/NC_005282]. S16: GenBank: NC_020416 [https://www.ncbi.nlm.nih.gov/nuccore/NC_020416].

### Cell-free transcription-translation
Cell-free gene expression was carried out using an *E. coli* TXTL system described previously[25,41] with one modification. We used the strain BL21-Δ*recBCD* Rosetta2 in which the *recBCD* gene set is knocked out to

prevent the degradation of linear DNA[68]. The preparation and usage of the TXTL system were the same as reported before[19,34]. Briefly, *E. coli* cells were grown in a 2xYT medium supplemented with phosphates. Cells were pelleted, washed, and lysed with a cell press. After centrifugation, the supernatant was recovered and preincubated at 37 °C for 80 min. After a second centrifugation step, the supernatant was dialyzed for 3 h at 4 °C. After a final spin-down, the supernatant was aliquoted and stored at −80 °C. The TXTL reactions comprised the cell lysate, the energy and amino acid mixtures, maltodextrin (30 mM) and ribose (30 mM), magnesium (2–5 mM) and potassium (50–100 mM), PEG8000 (3-4%), water and the DNA to be expressed. The reactions were incubated at 29 °C, in either 1.5 ml tubes or on 96 well plates (Plate reader H1m, Agilent, software Gen 5). For phage titration, the TXTL reactions were diluted with Luria broth (LB).

## DNA amplification
The Q5 high-fidelity PCR polymerase (NEB #M0491) was used to amplify the fragment tail fiber fragment E0 (gene *17* 1146−1662, gp17 382−554). PCR mutagenesis was carried out with the Agilent Gene-morph II Random mutagenesis Kit (#200550) according to manufacturer instructions. Low (E1), medium (E2), and high (E3) mutation rates were obtained by adding respectively 500 ng, 50 ng, and 0.5 ng of E0 in the initial PCR mix (50 μl) and performing 30 PCR cycles. The PCR fragments for PHEIGES assembly were otherwise amplified with KOD OneTM PCR Master Mix (No. KMM-101) according to manufacturer instructions. Either 1 μL of 1 ng/μL of T7 genome DNA or 1 μL of $10^5$ PFU/ml of clarified phage lysate obtained from a single plaque was used as template DNA. All PCR reactions were purified with a PCR clean-up kit (Invitrogen™ K310001) and DNA concentration was normalized to 20 nM in deionized water.

## DNA assembly and TXTL reactions
In vitro DNA assembly was adapted from a published protocol[33]. Briefly, purified PCR fragments were mixed at an equimolar concentration of 0.5-5 nM. An equal volume of a 2x assembly mix (20 mM Tris-HCl pH 7.9, 100 mM NaCl, 20 mM MgCl2, 10% (w/v) PEG8000, 2 mM Dithiothreitol, 1 U/μL Exonuclease III) and 2.5 μL DNA mix were mixed on ice. The tube containing the assembly mixture was transferred from ice to a water bath at 75 °C and incubated for 1 min. After 1 min at 75 °C, the tube was placed at room temperature and incubated for 5 min to anneal the DNA fragments with single-stranded ends. The mix was directly added to a TXTL reaction. Typically, 2.5 μL of the assembly mix was added to 7.5 μL of TXTL reaction. Negative controls were done by mixing all the PCR fragments except one replaced by an equal volume of water. For example, T7 genomic DNA was amplified in four fragments of 10.7 kbp (*T7A*), 9.9 kbp (*T7B*), 10.1 kbp (*T7C*), and 9.4 kbp (*T7D*) using four sets of oligonucleotides with 50 bp overlapping sequences (Supplementary Data 2). The fragments were individually amplified by PCR and purified using standard procedures (Supplementary Fig. 5). For the PCR amplification, a few picograms of genomes or a few microliters of a clarified phage lysate are added directly as a template to the PCR mixture. The size of each fragment was verified by standard DNA gel electrophoresis and their concentration was measured by spectrophotometry. The fragments were mixed at an equimolar concentration in the nanomolar range, treated with the *E. coli* exonuclease III and annealed. The exonuclease, rapidly deactivated during the procedure, produces single-stranded gaps at the 3' termini of the overlapping fragments enabling the annealing of complementary fragments. 2.5 μl of T7 DNA assembly reaction were added to a 7.5 μl TXTL reaction to express the annealed genomes. T7 phage genomes assembled from four parts were confirmed by NGS (T7-4PCS, Supplementary Data 3).

## NGS
Sequencing of the mutagenic PCR fragments was done by NGS (Illumina). DNA samples were converted to Illumina sequencing libraries using Illumina's Truseq NanoDNA Sample Preparation Kit (Cat. # 20015964). During library creation, amplicon DNA was end-repaired with the adapters, and indexes were ligated to each sample. The libraries did not undergo any PCR cycling. The final library size distribution was validated using capillary electrophoresis and quantified using fluorimetry (PicoGreen) and Kapa q-PCR. Pooled libraries were denatured and diluted to the appropriate clustering concentration. The libraries were then loaded onto the MiSeq paired-end flow cell and clustering occurred onboard the instrument. Once clustering was complete, the sequencing reaction immediately began using Illumina's 4-color SBS chemistry. Upon completion of read 1, 2 separate 8 or 10 base pair index reads were performed. Finally, the clustered library fragments were re-synthesized in the reverse direction thus producing the template for paired end read 2. Base call (.bcl) files for each cycle of sequencing were generated by Illumina Real Time Analysis (RTA) software. Primary analysis and de-multiplexing were performed using Illumina's bcl2fastq software 2.20. The result of the bcl2fastq workflow was de-multiplexed in FASTQ files. Reference mapping of the obtained reads and variant calling were done with open-source Galaxy software (BWA-MEM and iVar, and quality $Q \geq 30$ were used). Nucleotide mutations were analyzed with a Python program.

## Sanger and nanopore sequencing
Mutations in the T7 phage tail fiber gene *17* were analyzed by Sanger sequencing. Whole genome sequencing was performed by long-read sequencing technology from Oxford Nanopore Technologies service. DNA alignments for Sanger and Nanopore sequencing were performed on Benchling and analysis with a Python program.

## Phage spotting assay
1.5% agar-LB plates were pre-incubated at 37 °C for 1 h. 10 mL of 0.7% soft agar was kept at 55 °C in a water bath. 100 μL of overnight bacterial culture were mixed with the soft agar and vortexed gently. The soft agar was slowly dispensed onto the agar LB plates plate to cover uniformly the entire surface of the agar plate. The soft-agar plates were left at room temperature for 15 min on a flat surface to solidify. Serial ten-fold dilutions in LB of either cell-free phage reaction or clarified phage lysates were prepared in 200 μL. Spotting: for each phage dilution, 3.5 μL were dropped onto the soft agar. For negative control TXTL reactions, the whole reaction was diluted in LB at a final volume of 25 μL and spotted in one droplet onto the soft agar layer. After spotting, the plate was left for 15 min on the bench to let droplets absorb onto the soft agar. The plates were incubated at 37 °C, facing down, for 4 h. Plaques were counted at the dilution where 1 <#plaques <20 per spot. Titers were typically calculated from three serial dilution spotting. Plaque uncertainties are estimated at the time of counting (duplication or plaque belonging to the same spot). For calculations on PFUs, uncertainties were propagated using Python module *Uncertainties: a Python package for calculations with uncertainties*, Eric O. Lebigot.

## Phage infection kinetic assay
Infection kinetics were carried out in 96 well plates in a Synergy H1m microplate reader (Agilent). In each well, 180 μL of the host culture in LB (initial OD600 0.01-0.04) were mixed with 20 μL of different serial phage dilutions. Each condition was replicated in four different wells. Positive controls consisted of 180 μL hosts in LB + 20 μL LB. Optical density at 600 nm and fluorescence intensity (excitation 580 nm, emission 610 nm for mCherry) was blanked against wells containing 200 μL of LB at each timestep. A lid was added to the 96 well plates to reduce evaporation during acquisition. The microplate reader was set to 37 °C with continuous double orbital shaking at 200 rpm. Optical density and fluorescence intensity were measured in each well every 3 min during 5–10 h. The mean and standard deviation of each condition were calculated at each timestep.

## Microscopy

100 μL of 2% agar were poured onto a slide with a spacer, covered with a cover slip, and let to solidify at 4 °C 1 h. 2 μL of host cells (OD600 - 0.2) and 2 μL phage lysate ~10⁵ PFU/ml were added on the agar pad. The agar slide was incubated at 37 °C for 15 min to allow cells to decant on the agar layer. A slide was added on top of the agar layer and cell growth and lysis were recorded on an epifluorescence microscope (40x objective). Time Lapse movies were assembled by recording one image every 2 min at 37 °C using the software MetaMorph (MetaMorph Inc). Image composites and sequences were obtained with Fiji (https://imagej.net/software/fiji/).

## LPS-phage in vitro assay

LPS stock solutions were prepared in deionized water at 1 mg/ml and sonicated at 60 °C 30 min. Two conditions were used in this work: mild conditions (0.2 mg/mL, 2 h, 37 °C), stringent conditions (0.4 mg/mL, overnight, 37 °C). T7 clarified lysates ($10^7$–$10^{10}$ PFU/ml) were mixed with LPS (final 200 μg/ml) in a final 50 μL volume and incubated at 37 °C 2 h (mild conditions, Supplementary Figs. 15, 29). The phage LPS mixtures were serially diluted in LB and spotted on *E. coli* B lawn. For the p/G experiment, ReLPS were used in stringent conditions at 400 μg/mL and incubated at 37 °C overnight inhibit potential remaining ReLPS+ phage phenotypes (Supplementary Fig. 17).

## Statistics and reproducibility

For each experiment at least three independent replicates were made. Plaque uncertainties were estimated at the time of counting (duplication or plaque belonging to the same spot). For calculations of PFUs, uncertainties were propagated using Python module *Uncertainties: a Python package for calculations with uncertainties*, Eric O. Lebigot. The experiments were not randomized. The Investigators were not blinded to allocation during experiments and outcome assessment.

## Reporting summary

Further information on research design is available in the Nature Portfolio Reporting Summary linked to this article.

## Data availability

Source Data are provided with this paper. The data set used in the main Figures are presented as a Source data excel document. All Source Data necessary to reproduce analysis and plots of this paper are also available on the GitHub repository: https://github.com/Noireauxlab-TXTL/PHEIGES. Supplementary Source Data containing Source Microscopy Data are provided on the University of Minnesota Digital Conservancy (https://conservancy.umn.edu/) with the permanent URL: https://hdl.handle.net/11299/260237, and under public available license. The engineered T7 genome sequences generated in this study are available in the GenBank database under consecutive accession codes PP384393 to PP384410 (correspondence table provided in Supplementary Data 6). The mutated tip of tail fiber sequences generated in this study are available in the GenBank database under consecutive accession codes PP379475 to PP379532 (correspondence table provided in Supplementary Data 6). The raw genome data generated in this study are available in the SRA database under accession code PRJNA1077253. The raw NGS tail fiber data used in this study are available in the SRA database under accession code PRJNA1077490. Source data are provided with this paper.

## Code availability

The data analysis in this study was performed with Python codes, which are available on GitHub: https://github.com/Noireauxlab-TXTL/PHEIGES.

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

## Acknowledgements

The authors thank David Garenne and Seth Thompson for their help in the preparation of the TXTL system used in this work, Jaap Bosma and Irem Iskender for preparing salmonella phage genomes works. This work and the materials are based on funding provided by the National Science

Foundation (CBET FMRG 2228971). A.L. and I.K. gratefully acknowledges financial support for this publication by the Fulbright U.S. Student Program, which is sponsored by the U.S. Department of State, the Romanian-U.S. Fulbright Commission, and the Franco-American Fulbright Commission. We thank the University of Minnesota Genomic Center for NGS and Sorin Dinu for his kind help with NGS data analysis. A.L and A.B.L. thanks Ecole Doctorale Frontières de l'Innovation en Recherche et Education—Programme Bettencourt and the Bettencourt Schueller Foundation for their generous support.

## Author contributions

A.L. and I.K. performed the experiments. A.L., A.B.L and V.N. designed the experiments, analyzed the data, and wrote the manuscript. S.B. and B.N. edited the manuscript.

## Competing interests

A.L., A.B.L. and V.N. have submitted a patent application to the European Patent Office pertaining to the cell-free synthetis of phages of this work (application number EP23306788.3). A.L., A.B.L. and V.N. have submitted a patent application to the European Patent Office pertaining to the methods for making and screening a library of bacteriophages of this work (application number EP23306798.2). The remaining authors declare no competing interests.
