## [Peer Review File · Nature Communications]

REVIEWER COMMENTS

Reviewer #1 (Remarks to the Author):

The manuscript by Noireaux and colleagues demonstrates an efficient method to engineer the T7 phage in a cell-free extract. The method allows the introduction of insertions, deletions, and point mutations simultaneously (multiplexing). The authors further demonstrate that if two genomes or more are mixed in the cell-free extract, there is a higher probability that proteins produced from each genomic DNA will be used to assemble the capsid encapsulating this same genomic DNA. This linkage between the DNA and the proteins produced by it allows selection of certain traits such as ligand specificity to the receptor, which the authors demonstrate.

The manuscript is clearly presented in most parts. It becomes complicated in the parts describing the expected versus observed probabilities of the linkage of the genotype to the phenotype. The figures make the complicated parts easier to understand. Supplementary data is sufficient, and the major data is shown in the main figures.

Below are some of the concerns.

1. While the authors nicely demonstrate the phenotype-genotype linkage in their system, the results derived from the assays do not provide new information, but are rather based on the known literature.

A possible way to show novel results is to use the established T7-phage display system. Can the authors use the principles of this system to fuse to Gp10 a library of peptides, some of which are toxic to *E. coli*, and thus cannot be used in the original phage-display system, and then use panning to identify a toxic peptide binding its receptor? This will demonstrate the advantage of the system over regular phage-display system. It will also demonstrate that the linkage is robust between the genotype and the phenotype. This will also demonstrate the strength of the approach in binding and not only in harsher selection like DNA ejection caused by LPS.

2. The authors claim that the TXTL can be easily applied to coliphages, and also to other phages in the presence of specific transcription factors, citing their previous work. However, some phages require host factors for their propagation. Absence of thioredoxin from the extract, for example, would not allow T7 growth. Such factors may be required for many other non-coliphages, and thus the system may not be so general.

3. Synthesis of large phages such as T6 is cumbersome, requiring more than a dozen pieces of PCRs. The engineering system is therefore, in my opinion, not suitable for such purposes, where much better alternatives are available.

Minor issues:

1. Line numbers would be helpful to point out specific concerns better.

2. Ando et al. reference is missing – when mentioning phage rebooting in yeast, and in other places regarding tail mixing and replacement - [https://www.cell.com/cell-systems/pdfExtended/S2405-4712\(15\)00111-8](https://www.cell.com/cell-systems/pdfExtended/S2405-4712(15)00111-8).

3. Page 6 “in vitro used for infection” – unclear sentence.

Reviewer #2 (Remarks to the Author):

The authors described the PHEIGES system, which combines an in vitro DNA assembly method using exonuclease III and the modified TXTL toolbox 3.0. After DNA assembly, the reactant can be conveniently taken directly to the TXTL reaction without purification. In addition, by applying orthogonal primers, the authors showed zero-background leak-free PHEIGES that do not allow rebooting of non-edited WT phages. All experiments except Figure S31 used the model coliphage T7 and generated WT phage and various genetically modified phages (mCherry-loaded reporter phage, phage that can infect LPS mutants, phage with ca. 10% reduced genome, or a combination of these) from the leak-free PHEIGES. Moreover, the characteristics of the TXTL system, i.e., different designer genomes can co-exist and be expressed simultaneously in the system, were used to create T7 phages with a combination of different genomes and chassis without linking genotype and phenotype. The authors also applied this to the evaluation of randomness of tail fiber assembly in the PHEIGES. Although no genotype-phenotype coupling was predicted for the TXTL system, the experimental results suggest that a coupling exists, leading to the interesting observation that this coupling might arise due to an unknown interaction between phage proteins and their genome, or the specific circumstances of a TXTL reaction.

This manuscript is written well and scientifically sound. The leak-free PHEIGES system is interesting, and it appears to be useful, at least for modification of T7 phage. However, the system is a combination of already established methods, namely the in vitro DNA assembly using exonuclease III reported by Nozaki, TXTL toolbox 3.0 reported by the authors' group, and orthogonal primers design by Subramanian et al. Moreover, the combination of cell-free genome assembly and cell-free rebooting has previously been reported, with both genome size reduction and the insertion of reporter genes achieved through a comparable approach. The experimental setup of genotype-phenotype coupling for T7 phage using the TXTL system is intriguing; however, unexpected results contrary to the authors' expectations have been obtained, and the discussion regarding these outcomes is insufficient. Additionally, except for genotype-phenotype coupling experiments, the use of leak-free PHEIGES is essentially limited to seeing similar results reported previously. Demonstrating non-model and/or non-coliphage engineering is very important for this manuscript, and thus the authors can strengthen the versatility and accessibility of PHEIGES.

Comments

(1) Figure S1 bottom right: If the authors would like to show that bacteria grow when there is contamination in the TXTL sample, it would be more appropriate to use *E. coli* BL21ΔrecBCD Rosetta 2, which was used for TXTL preparation.

(2) Figure S6d-f, S9d, S19 bottom : Please place a positive control.

(3) Pages 8-9: No Figure 4f (bottom right?) and its legend.

(4) The parent strain of Keio LPS mutants is *E. coli* BW25113 which is a derivative of *E. coli* K-12, not B. The authors should use BW25113 as the parent strain for spot assays and related experiments or construct LPS mutants from *E. coli* B.

(5) In relation to the comment above, for example in Figures S14 and S21, the EOP of the strain B is set to 1. The EOP of BW25113 should be set to 1 or it should be shown that there is no difference between the EOP of strains B and BW25113.

(6) Related to the comment above, which strain was used for propagation of T7 variants that can infect *E. coli* LPS mutant? Does the bacterial modification-restriction system affect EOP when the host for phage propagation and the host for spot assays are different? This is another reason why I asked Comment (4).

(7) Figure S12: Please clarify about the LPS type of strain B. In the table, described as analogue to RaLPS. In the figure legend, described as RbLPS.

(8) Figure S15 and S29: KLA (Kdo2-Lipid A) is only mentioned in Methods section. It would be clearer to describe it as ReLPS instead of KLA.

(9) Figure S31: Any reason the authors used plaque formation assay and spotting assay differently?

(10) Figure S31: Please add provider's information and experimental conditions.

(11) The description of genotype-phenotype linkage was difficult to follow, at least for me. I recommend creating a Table(s) to show the list of numbers described in the supplementary figure legend for the results of the spot assay and revising the text.

(12) It would be appreciated if the authors would add line numbers to make it easier to comment.

Page 2: In this regard, the CFS... The cell-free synthesis (CFS) of phages... → In this regard, the cell-free synthesis (CFS)... The CFS of phages... .

Figure 1 legend: CFE → CFS

Page 3: We then re-assembled the T7 WT phage (Fig. S5) from four PCR amplified fragments (Fig 2a) with... → We then re-assembled the T7 WT phage (Fig. 2a) from four PCR amplified fragments (Fig. S5) with...

Page 3: Five gene fragments were amplified using five sets of... → Six gene fragments were amplified using six sets of...

Figure 6 legend: RePLS → ReLPS

Figure S7 legend: the five DNA fragments... the four DNA fragments... → the six DNA fragments... the five DNA fragments...

Reviewer #3 (Remarks to the Author):

Title: PHEIGES, all-cell-free phage synthesis and selection from engineered genomes

General Comment:

Levrier et al. performed elegant studies demonstrating that genotype-phenotype linkage of engineered phages can occur in 'rebooted' phage from cell free extract. Because of this new found property, the authors were able to easily and quickly make T7 tail fiber libraries to alter T7 host range to infect known T7 resistant mutants. In addition, the authors show that by using a newly developed gene assembly method, they were able to improve the yield of phage rebooting, require less DNA, and streamline the process to be easier because DNA purification of large phage genomes is not required. Additionally, the authors also highlighted the utility of their method of cell-free synthesis of engineered phages by engineering a minimal T7 phage and also a reporter phage encoding a fluorescent protein reporter.

Because of the unique and unexpected properties of g/p linkage and the growing public health problem of antibiotic resistance, I believe this work will be of interest to the broad audience of Nature Communications.

Specific Comments (not in order of importance, but rather in order of text):

1. Define CFS when first using.
2. The Cell-free transcription-translation results discussion seems out of place and not necessary.
3. How long was DNA incubated with exonuclease? This information isn't described anywhere in the text and seems like it would be very important to replicate the results. Do the authors think there is an optimal time?
4. Since gene assembly is very important and is described as one of the novelties of the approach, can the authors please include more details in figure 1. Like addition of needing an exonuclease and then hybridization.
5. The authors mention this approach takes a day. How long do other comparable methods take?
6. Can the authors give more detail on the RFP selection? For example, were a thousand single plaques picked and grown in a 96 well plate and screened for fluorescence because of the leaky assembly? I could not easily find this information.
7. Can the authors give more detail of Figure S7. Are each row a replicate or individual experiment?
8. Can the authors give more details on the 'orthogonal oligonucleotide/sense and antisense' strategy. Inherent to the gene assembly design, aren't all the fragments orthogonal? This seems very important for improving the yield, yet it isn't clearly described what this strategy is.
9. 'The plaques formed by T7-mini are clear and circular (Fig. 3c):' : They look less clear to me.

10. How was this estimated:

We estimated that the T7-rfaD-1 variant is at least ~10,000 times more sensitive to ReLPS in vitro than T7 WT.

11. Please reference SI discussion about assumptions, when describing the original assumption. I found this discussion very interesting, yet it is buried in a reference in the main text. Also, to differentiate between a mixture and 100% MUT required for infectivity, couldn't the authors passage their libraries once and measure the % composition change.

12. Which is what: ($1.9 \pm 0.4 \times 10^6$, $1.7 \pm 0.4 \times 10^7$ PFU/mL, respectively).

13. How were these conditions different than the overnight. i.e., how long is extended period and what was the LPS concentration for the 50% mixture:

Separately assembled T7-vWT phages were poorly susceptible to ReLPS when incubated with 0.4 mg/mL ReLPS for an extended period at 37 °C (10-fold decrease of titer from $\sim 6 \times 10^8$ PFU/mL to $\sim 5 \times 10^7$ PFU/mL).

14. what % is this and was this accounted for in the calculations:

A few undefined plaques were also detected on rfaC from the equal volume mix of pre-assembled T7-vWT and -rfaD-1 phages after the ReLPS ejection and amplification on WT host.

15. Can the authors introduce and describe the (i) and (ii) in better detail. Seem abruptly introduced.

16. What is 'pure and opposing p/Gs'?

17. Can the authors please elaborate why H2 was chosen for the hypothetical plot for random mix in Figure 4e or 4f?. The authors begin the discussion assuming only one monomer is required for phage infection. Isn't that H0? Also, would be interesting to model if 1 monomers, 2 monomers, 3 monomers... are required for infectivity:

We calculated the theoretical fully randomized relative abundance of the six different g/P

hybrids under the most permissive assumption, namely that the integration of at least one S541R mutant monomer into one of the six trimeric tail fibers of a given phage is necessary and sufficient to infect rfaC ReLPS (Fig. 4d).

18. How does the % composition change if the authors grow the 50% mixture for a single infection cycle on E. coli B and then plaque on E. coli B and ReLPS, essentially converting all the mut/Mix to mut/MUT and wt/Mix to wt/WT? This experiment should differentiate whether one tail fiber of MUT is needed or if 100% MUT is needed. (same comment as 11b).

19. Please include DOI: 10.1016/j.cell.2019.09.015 here: 'The tip of the tail fiber gene 17 (amino acids 472-554), determinant of the phage's host range^{53 54 48}, can be exchanged between phages or mutated to adapt to new hosts^{55 56 57 58}.'

20. Please specific in the figure what library each phage came from:

Twelve clonal phages from each of the three hosts (Fig. 5d-f) were purified, propagated in their respective host, phenotyped on all hosts (Fig. S22, S23, S24) and sequenced (Fig. S25).

21. The authors should consider including this data in main text as a bar graph. Just the C, E, A mutants: This strongly supports a g/P linkage in PHEIGES. Interestingly, titers were somewhat lower on the rfaC as compared to the rfaE or lpcA host strains although they are believed to share the same ReLPS serotype. This may suggest different membrane properties between the knockouts.

22. Discussion is severely lacking. Please expand the discussion to describe how this library generation approach compares in efficiency of expanding phage host range to other methods and the nuances between each.

23. If g/P weren't coupled, couldn't one simply add a single passage step, to make all the libraries mut/mut? Could the authors please include this in the discussion, and potential biases of this approach?

Reviewer #1 (Remarks to the Author):

The manuscript by Noireaux and colleagues demonstrates an efficient method to engineer the T7 phage in a cell-free extract. The method allows the introduction of insertions, deletions, and point mutations simultaneously (multiplexing). The authors further demonstrate that if two genomes or more are mixed in the cell-free extract, there is a higher probability that proteins produced from each genomic DNA will be used to assemble the capsid encapsulating this same genomic DNA. This linkage between the DNA and the proteins produced by it allows the selection of certain traits such as ligand specificity to the receptor, which the authors demonstrate.

The manuscript is clearly presented in most parts. It becomes complicated in the parts describing the expected versus observed probabilities of the linkage of the genotype to the phenotype.

We thank the reviewer for his overall appreciation of our work and suggestions listed below with our intercalated responses. We further clarified the manuscript text concerning the different hypotheses underlying expected probability and their underlying assumptions vs. the observable results (p. 8 lines 283-289):

'Equations (1) and (2) describe the assumptions that the ratio of pure p/G phages (mut/MUT and wt/WT) as well as opposing p/G (mut/WT and wt/MUT) on all the total number of phages that carry a mut or a wt genotype are the same. We define pure p/G as a phage that displays only tail fibers that correspond to its genomic sequence, for example, wt genotype and only WT tail fibers. Conversely, we define as opposing p/G a phage that displays only tail fibers that do not correspond to the genomic sequence for example wt genotype and only MUT tail fibers.'

The figures make the complicated parts easier to understand. Supplementary data is sufficient, and the major data is shown in the main figures.

Below are some of the concerns.

1. While the authors nicely demonstrate the phenotype-genotype linkage in their system,

Indeed, demonstrating T7 gp17 phenotype-genotype linkage has never been reported before and can lead to further future novel insights into viral assembly inasmuch as to biotechnological advances (e.g. phage display).

the results derived from the assays do not provide new information but are rather based on the known literature. A possible way to show novel results is to use the established T7-phage display system. Can the authors use the principles of this system to fuse to Gp10 a library of peptides, some of which are toxic to E. coli, and thus cannot be used in the original phage-display system, and then use panning to identify a toxic peptide binding its receptor? This will demonstrate the advantage of the system over the regular phage-display system. It will also demonstrate that the linkage is robust between the genotype and the phenotype. This will also demonstrate the strength of the approach in binding and not only in harsher selection like DNA ejection caused by LPS.

The reviewer asserts that ‘While the authors nicely demonstrate the phenotype-genotype linkage in their system, the results derived from the assays do not provide new information but are rather based on the known literature’. We wish to note that the claim of no novelty and basis in known literature is incorrect. All the extensive literature thus far concerning phage display (and actually for all directed evolution systems as a whole) depends on compartmentalization to achieve genotype-phenotype linkage, either within cells or synthetic compartments (*i.e.*, droplets, achieved in microfluidics or emulsion systems). The only current methodology dispensed of a compartmentalization solution is Ribosome Display. The latter is limited by the fragility of the mRNA-ribosome-polypeptide linkage and the fragility of the mRNA. We now highlighted this in the main text (p. 7 Lines 223-227):

‘In laboratory settings, genetic phage variant libraries are either directly transformed into host cells, or they are first packaged *in vitro* and then used to infect host cells to reveal their linked phenotypes. The only current methodology dispensed of a compartmentalization is Ribosome Display. The latter is limited by the fragility of the mRNA-ribosome-polypeptide linkage and the fragility of mRNA.’

Thus, we deem that the surprising result of achieving phenotype-genotype linkage in PHEIGES is novel and worthy of publication in this journal.

We believe that further demonstration of a fully-fledged phage display to select for novel function as suggested by the reviewer (*e.g.*, selection of peptides that otherwise would be toxic to the *E. coli* host) is beyond the scope of this paper. It would also need a dedicated 6-12 months of work to establish a separate high-impact manuscript.

However (!), whether PHEIGES can be used to successfully assemble protein fusion to either the tail protein (gp17) or to the major capsid protein (gp10) is an intriguing and important question that was not addressed in the original manuscript. Demonstrating such capacity would indeed serve as the basis for further developing phage display and to examine whether the genotype-phenotype linkage observed for the tail protein (gp17) could be extended to the major capsid protein (gp10). To this end, we added new experimental results demonstrating that using PHEIGES, 3xFlag tag peptide (22 amino acids) can be fused to either of the above phage proteins and assembled, resulting in assembled phages as described in p. 13 Lines 427-433 and Fig. S28.

‘Peptide display using PHEIGES. In order to pursue in the future *in vitro* selection based on PHEIGES, we tested whether PHEIGES could successfully assemble phages with peptide fusions to either the main coat protein (gp10b) as is commonly practiced⁴⁷ or to the tail fiber protein (gp17). To this end, we used as a model a triple repeat of the FLAG tag peptide (DYDDDDK). Our results (Fig. S28) demonstrate successful phage assembly of both gp10b and gp17 FLAG fusions. Notably, T7 phages displaying gp17-FLAGx3 fusion, retained their infectibility, suggesting the fusion does not hinder the interaction between the tail fiber and the LPS host receptor.’

As expected, packaging of translational fusions of larger polypeptides (e.g. GFP; 227 amino acids) was unsuccessful, due to packaging constraints.

2. The authors claim that the TXTL can be easily applied to coliphages, and also to other phages in the presence of specific transcription factors, citing their previous work. However, some phages require host factors for their propagation. The absence of thioredoxin from the extract, for example, would not allow T7 growth. Such factors may be required for many other non-coliphages, and thus the system may not be so general.

The absence of thioredoxin from the extract does not prevent T7 synthesis in cell-free systems as demonstrated by us in our article <https://pubmed.ncbi.nlm.nih.gov/23651338/>. It is a surprising result, but all the controls were done correctly, and this experiment was repeated several times. Thioredoxin is important but not essential, at least in cell-free. Our original phrasing was cautious as we do not claim that PHEIGES is easy to apply to all coliphages. We claim that PHEIGES may be applied to infectious phages that have been proven to be cell-free synthesized in our TXTL system, up to genome sizes of 169 kbp (T4 and T6 phages). This may include other phages (Simmel and co-workers (Cell-free production of personalized therapeutic phages targeting multidrug-resistant bacteria - PubMed (nih.gov)) added to the revision: T4 (Fig S32), (~88 kbp) T6 (~170 kbp) along with Salmonella phages FelixO1 (~87 kbp) and S16 (~169 kbp). (Fig. S33). In additional support, we provide further new experimental results demonstrating genome re-assembly and cell-free synthesis of the FelixO1 salmonella phage (86 kbp). This is referred to on p. 14 Lines 451-462 and added figures (SI figure S22 and S33):

‘Towards non-model phages. This work is based on the model T7 phages to establish PHEIGES workflow. T4 phage (~169 kbp) was previously expressed from its purified genomes in *E. coli* TXTL. Subsequently, we reproduced T4 TXTL expression (Fig S32) and, for the first time, demonstrated the expression of *E. coli* phages VpaE1 (~88 kbp) T6 (~170 kbp) along with Salmonella phages FelixO1 (~87 kbp) and S16 (~169 kbp) from their purified genomes at 0.1 nM in TXTL. These results suggest the expression capability for a spectrum of *E. coli* and non-*E. coli* phages with genomes ranging from 40 kbp to 170 kbp in *E. coli*-based TXTL. To assess scalability, we applied PHEIGES to Salmonella phage FelixO1. We re-assembled FelixO1 genome from five PCR fragments directly amplified from a salmonella phage lysate. We obtained ~10⁸ PFU/mL of FelixO1 phages (fig S33, table S1, S2). This result suggests that PHEIGES could be applied to engineer a range of non-*E. coli* phages with larger genomes than T7. To our knowledge, FelixO1 is the largest phage genome rebooted *in vitro*.

3. Synthesis of large phages such as T6 is cumbersome, requiring more than a dozen pieces of PCRs. The engineering system is therefore, in my opinion, not suitable for such purposes, where much better alternatives are available.

We admit we differ in our opinion as dozens of PCR can be performed in parallel, with no additional time and at negligible cost increase. However, we agree that in the case of T6

and T4 genome DNA modifications complexify in vitro approaches like PHEIGES. We now refer to this in the main text (p. 14 lines 462 – 465):

Ongoing investigations will elucidate the applicability of PHEIGES for engineering larger phages with chemically modified genomes like T4 and T6 phages or non-*E. coli* phages that may require alternative expression platforms beyond *E. coli*-based TXTL.'

Minor issues:

1. Line numbers would be helpful to point out specific concerns better.

We apologize for this neglect and now added line numbers.

2. Ando et al. reference is missing – when mentioning phage rebooting in yeast, and in other places regarding tail mixing and replacement.

Thank you for the suggestion, indeed relevant! The reference was added twice at the indicated locations in the manuscript (p. 2, line. 39 and p. 10 line 325)
<https://pubmed.ncbi.nlm.nih.gov/26973885/>

3. Page 6 “in vitro used for infection” – unclear sentence.

The sentence now reads as (p. 7 Line 223):

'In laboratory settings, genetic phage variant libraries are either directly transformed into host cells, or they are first packaged in vitro and then used to infect host cells to reveal their linked phenotypes.'

Reviewer #2 (Remarks to the Author):

The authors described the PHEIGES system, which combines an *in vitro* DNA assembly method using exonuclease III and the modified TXTL toolbox 3.0. After DNA assembly, the reactant can be conveniently taken directly to the TXTL reaction without purification. In addition, by applying orthogonal primers, the authors showed zero-background leak-free PHEIGES that do not allow rebooting of non-edited WT phages. All experiments except Figure S31 used the model coliphage T7 and generated WT phage and various genetically modified phages (*mCherry*-loaded reporter phage, phage that can infect LPS mutants, phage with ca. 10% reduced genome, or a combination of these) from the leak-free PHEIGES. Moreover, the characteristics of the TXTL system, i.e., different designer genomes can co-exist and be expressed simultaneously in the system, were used to create T7 phages with a combination of different genomes and chassis without linking genotype and phenotype. The authors also applied this to the evaluation of randomness of tail fiber assembly in the PHEIGES. Although no genotype-phenotype coupling was predicted for the TXTL system, the experimental results suggest that a coupling exists, leading to the interesting observation that this coupling might arise due to an unknown interaction between phage proteins and their genome, or the specific circumstances of a TXTL reaction. This manuscript is written well and scientifically sound. The leak-free PHEIGES system is interesting, and it appears to be useful, at least for modification of T7 phage. However, the system is a combination of already established methods, namely the *in vitro* DNA assembly using exonuclease III reported by Nozaki, TXTL toolbox 3.0 reported by the authors' group, and orthogonal primers design by Subramanian *et al.* Moreover, the combination of cell-free genome assembly and cell-free rebooting has previously been reported, with both genome size reduction and the insertion of reporter genes achieved through a comparable approach. The experimental setup of genotype-phenotype coupling for T7 phage using the TXTL system is intriguing; however, unexpected results contrary to the authors' expectations have been obtained, and the discussion regarding these outcomes is insufficient.

We appreciate the understandable Reviewer's difficulty and clarify further this section and added the actual titers (see point #11 below).

Additionally, except for genotype-phenotype coupling experiments, the use of leak-free PHEIGES is essentially limited to seeing similar results reported previously. Demonstrating non-model and/or non-coliphage engineering is very important for this manuscript, and thus the authors can strengthen the versatility and accessibility of PHEIGES.

We thank the author for the suggestion. We now added a new section to describe new experiments to demonstrate the assembly of FelixO1, a *Salmonella* (non-model, non-coli) phage, genome and phage synthesis using PHEIGES (Fig, S33). Felix-O1 genome size (86 kbp) is more than the double of T7. Yet we also probed the limitations of PHEIGES, showing that, similarly to *in vivo* approaches such as CRISPR, it currently cannot handle large phage genomes that are chemically modified (such as T4 and T6). Text p. 14 Line 541-464):

Towards non-model phages. This work is based on the model T7 phages to establish PHEIGES workflow. T4 phage (~169 kbp) was previously expressed from its purified genomes in *E. coli* TXTL. Subsequently, we reproduced T4 TXTL expression (Fig S32) and, for the first time, demonstrated the expression of *E. coli* phages VpaE1 (~88 kbp) T6 (~170 kbp) along with Salmonella phages FelixO1 (~87 kbp) and S16 (~169 kbp) from their purified genomes at 0.1 nM in TXTL. These results suggest the expression capability for a spectrum of *E. coli* and non-*E. coli* phages with genomes ranging from 40 kbp to 170 kbp in *E. coli*-based TXTL. To assess scalability, we applied PHEIGES to Salmonella phage FelixO1. We re-assembled FelixO1 genome from five PCR fragments directly amplified from a salmonella phage lysate. We obtained ~10⁸ PFU/mL of FelixO1 phages (fig S33, table S1, S2). This result suggests that PHEIGES could be applied to engineer a range of non-*E. coli* phages with larger genomes than T7. To our knowledge, FelixO1 is the largest phage genome rebooted *in vitro*. Ongoing investigations will elucidate the applicability of PHEIGES for engineering larger phages with chemically modified genomes like T4 and T6 phages or non-*E. coli* phages that may require alternative expression platforms beyond *E. coli*-based TXTL.

Comments

(1) Figure S1 bottom right: If the authors would like to show that bacteria grow when there is contamination in the TXTL sample, it would be more appropriate to use *E. coli* BL21ΔrecBCD Rosetta 2, which was used for TXTL preparation.

We followed this suggestion and updated SI Figure1 accordingly. The legend now reads:

'Bottom right: as a positive control, *E. coli* BL21 ΔrecBCD Rosetta 2 cells were added to a TXTL sample to show that cells are viable in TXTL.'

(2) Figure S6d-f, S9d, S19 bottom : Please place a positive control.

We purposely omitted a positive control on S6d-f plate to prevent potential contaminations from the positive control on the other spottings. The positive controls are in S6a-c. Same for S9d, the positive controls are in S9a-c. Same for S19, the positive controls are just above to avoid potential contaminations. Note that the positive control, which consists of assembling the phage T7 genome from 4 fragments, was also done routinely throughout our work. In all cases, the positive controls were made on bacterial loans plated in the same day and from the same batch.

(3) Pages 8-9: No Figure 4f (bottom right?) and its legend.

We apologize for our mistake; we have now changed the reference to Fig. 4f with Fig. 4e. Figure 4f does not exist.

(4) The parent strain of Keio LPS mutants is *E. coli* BW25113 which is a derivative of *E. coli* K-12, not B. The authors should use BW25113 as the parent strain for spot assays and related experiments or construct LPS mutants from *E. coli* B.

(5) In relation to the comment above, for example in Figures S14 and S21, the EOP of the strain B is set to 1. The EOP of BW25113 should be set to 1 or it should be shown that there is no difference between the EOP of strains B and BW25113.

(6) Related to the comment above, which strain was used for propagation of T7 variants that can infect *E. coli* LPS mutant? Does the bacterial modification-restriction system affect EOP when the host for phage propagation and the host for spot assays are different? This is another reason why I asked Comment (4).

Response to comments 4, 5, 6:

We now show that the titers of T7 WT amplified either from *E. coli* B, the BW25113 strain or from cell-free synthesis are the same on *E. coli* B and BW25113 (Fig. S14 b). Therefore, we updated the text p. 6 line 199 to indicate that the EOP of BW25113 was set to 1 (SI Fig. S14 b):

'T7 lysate from *E. coli* B, KEIO parent strain BW25113 and TXTL expressed. Similar titers are observed in all conditions. This suggests that *E. coli* B and K12 derivative strains modification-restriction systems do not affect T7 EOP and that T7 interacts with *E. coli* B and *E. coli* BW25113 in the same way.'

The fact that T7 WT on *E. coli* B or BW25113 have the same EOP indicates that the restriction modification system does not affect the EOP but that it is rather the T7 receptor (LPS) that affects the EOP.

Phages were not propagated prior to spotting. TXTL reactions were only diluted and spotted directly on the host of interest. Single plaques on the LPS host of interest were amplified on the same host and sequenced.

(7) Figure S12: Please clarify about the LPS type of strain B. In the table, described as analogue to RaLPS. In the figure legend, described as RbLPS.

We corrected this mistake (Fig. S14), *E. coli* strain B is an RbLPS, we added a reference that indicates this (<https://pubmed.ncbi.nlm.nih.gov/27273222/>).

(8) Figure S15 and S29: KLA (Kdo2-Lipid A) is only mentioned in Methods section. It would be clearer to describe it as ReLPS instead of KLA.

We thank the reviewer for pointing this out, we changed KLA to ReLPS (Fig S15 and S30). KLA is only mentioned in the methods section.

(9) Figure S31: Any reason the authors used plaque formation assay and spotting assay differently?

The assays are identical and no true reason to name them differently. These plating and spotting assays were made in two different labs (Bowden and Noireaux labs) due to some

being BSL1 and some BSL2 (FelixO1). We homogenized the manuscript to use plaque formation/spotting throughout the text.

(10) Figure S31: Please add provider's information and experimental conditions.

We added this information in renumbered Figure S32 legend and in the acknowledgments.

(11) The description of genotype-phenotype linkage was difficult to follow, at least for me. I recommend creating a Table(s) to show the list of numbers described in the supplementary figure legend for the results of the spot assay and revising the text.

Figures S16 and S17 were modified: the titers are now listed under each case in the figure and can be easily compared. The text of the legends has been revised accordingly.

(12) It would be appreciated if the authors would add line numbers to make it easier to comment.

Our apologies, we now added line numbers.

Page 2: In this regard, the CFS... The cell-free synthesis (CFS) of phages... → In this regard, the cell-free synthesis (CFS)... The CFS of phages... .

This was corrected.

Figure 1 legend: CFE → CFS

This typo was corrected.

Page 3: We then re-assembled the T7 WT phage (Fig. S5) from four PCR amplified fragments (Fig 2a) with... → We then re-assembled the T7 WT phage (Fig. 2a) from four PCR amplified fragments (Fig. S5) with...

This was corrected.

Page 3: Five gene fragments were amplified using five sets of... → Six gene fragments were amplified using six sets of...

This was corrected.

Figure 6 legend: RePLS → ReLPS

The typo was corrected.

Figure S7 legend: the five DNA fragments... the four DNA fragments... → the six DNA fragments... the five DNA fragments...

This was corrected.

Reviewer #3 (Remarks to the Author):

Title: PHEIGES, all-cell-free phage synthesis and selection from engineered genomes

General Comment:

Levrier et al. performed elegant studies demonstrating that genotype-phenotype linkage of engineered phages can occur in 'rebooted' phage from cell free extract. Because of this new found property, the authors were able to easily and quickly make T7 tail fiber libraries to alter T7 host range to infect known T7 resistant mutants. In addition, the authors show that by using a newly developed gene assembly method, they were able to improve the yield of phage rebooting, require less DNA, and streamline the process to be easier because DNA purification of large phage genomes is not required. Additionally, the authors also highlighted the utility of their method of cell-free synthesis of engineered phages by engineering a minimal T7 phage and a reporter phage encoding a fluorescent protein reporter. Because of the unique and unexpected properties of g/p linkage and the growing public health problem of antibiotic resistance, I believe this work will be of interest to the broad audience of Nature Communications.

Specific Comments (not in order of importance, but rather in order of text):

1. Define CFS when first using.

This was corrected.

2. The Cell-free transcription-translation results discussion seems out of place and not necessary.

Having the paragraph on TXTL at the beginning of the results section is essential to us as it sets critical and solid ground for the rest of the work; namely:

- Provide 'best practices' (often missing in the field) - We demonstrate that the TXTL used in our work does not contain any remaining cells by systematically testing each lysate for remaining live cells, which is critical for cell-free phage work. It is not an obvious point because many laboratories preparing their own TXTL systems do not test for this, surprisingly.
- We target broad audience who may not be familiar with TXTL system. For the experts, it is important to note that the TXTL used in this work is slightly different than usual as we use the same parent strain but with a recBCD knockout, to stabilize linear DNA.

We thus opted to keep this paragraph.

3. How long was DNA incubated with exonuclease? This information isn't described anywhere in the text and seems like it would be very important to replicate the results. Do the authors think there is an optimal time?

This was clarified in the method section p. 17 Line 574 as:

'The tube containing the assembly mixture was transferred from ice to a water bath at 75 °C and incubated for 1 min. After 1 min at 75 °C, the tube was placed at room temperature and incubated for 5 min to anneal the DNA fragments with single-stranded ends.'

We think indeed there is an optimal time that depends on factors such as the composition of the assembly buffer, DNA concentrations, length of the overlapping sequence and number of fragments.

4. Since gene assembly is very important and is described as one of the novelties of the approach, can the authors please include more details in figure 1. Like addition of needing an exonuclease and then hybridization.

Figure 1 was updated, and the legend adapted accordingly p. 3 line 77:

'The PCR products are cleaned up, briefly digested with a 3' exonuclease to create sticky ends and annealed in vitro.'

5. The authors mention this approach takes a day. How long do other comparable methods take?

We developed the discussion to compare our approach with others (p.14-15 Line 467-478):

'PHEIGES provides a rapid, technically accessible, and low-cost method for phage engineering. Its DNA assembly efficiency ($\sim 10^{10}$ PFU/ml) offers many advantages compared to the current in vivo methods^{15 15 18 19} and other cell-free phage engineering methods^{21 40 63}. The yeast cloning approach to phage genome engineering takes on the order of one week to achieve^{64 65}. CRISPR approaches take at least one week to achieve as plasmids must be prepared for homologous recombination first^{13 14}. The Golden Gate Assembly method (GGA) takes about two to three days to carry out without taking into account the step of whole genome synthesis to remove unwanted GGA sites⁶³. Taking into account the genome synthesis, the GGA method takes more than one week. The GGA method is also not compatible with TXTL, consequently the assembly reaction must be cleaned up first, a step that considerably reduces the yields⁶³, which is also the major limitation to the Gibson assembly method for phage engineering ($< 10^5$ PFU/ml)⁴⁰. The PHEIGES workflow presented in this work eliminates all these steps.'

6. Can the authors give more detail on the RFP selection? For example, were a thousand single plaques picked and grown in a 96 well plate and screened for fluorescence because of the leaky assembly? I could not easily find this information.

This is described in the legend of S8: 'A serial dilution of the TXTL reaction expressing and synthesizing the T7-mCherry phages was added to E. coli B cultures to isolate phages carrying the mcherry gene cassette. With a concentration one thousand greater than T7 WT, T7-mCherry phages were isolated by taking the last dilution that exhibited mCherry fluorescence.'

7. Can the authors give more detail of Figure S7. Are each row a replicate or individual experiment?

Figure S7 was clarified: 3 independent assembly experiments, each spotted on three rows.

8. Can the authors give more details on the 'orthogonal oligonucleotide/sense and antisense' strategy. Inherent to the gene assembly design, aren't all the fragments orthogonal? This seems very important for improving the yield, yet it isn't clearly described what this strategy is.

We added the following sentences in the main text to explain these differences (P 4 Line 138-142):

'In our first strategy, the overlapping sequences used for annealing the PCR fragments are the natural genomic sequences, which may not be strongly orthogonal to each other. Incorporating synthetic orthogonal sequences into the oligonucleotides used to generate the PCR fragments decreases potential crosstalk between overlapping sequences during annealing'.

9. 'The plaques formed by T7-mini are clear and circular (Fig. 3c).': They look less clear to me.

We updated Figure 3c with new images. There are no differences between the plaques. They may seem different on short incubation times (2h) because infection by T7 mini is slower than T7 WT as described in Fig. S29.

10. How was this estimated: We estimated that the T7-rfaD-1 variant is at least ~10,000 times more sensitive to ReLPS in vitro than T7 WT.

We thank the reviewer for highlighting this. We added the explanation in the legend (Fig. S15), this estimation corresponds to the ratio of $10^3 / (4 \times 10^7)$ for T7-rfaD-1, which is compared to the ratio of the T7WT which is $1 \times (10^8 / 10^8)$, which in the end corresponds to at least 10^4 . (Note that the 10^3 is actually $<10^3$):

'The titer loss of T7-rfaD-s compared to T7 WT in presence of ReLPS shows that T7-rfaD-1 is at least ~10 000 times $(T7WT-ReLPS/T7WT-water)/(T7ReLPS-ReLPS/T7ReLPS-water)$ more selective to ReLPS in vitro than T7 WT at 0.2 mg/mL ReLPS 37 °C, 2h.'

11. Please reference SI discussion about assumptions, when describing the original assumption. I found this discussion very interesting, yet it is buried in a reference in the main text. Also, to differentiate between a mixture and 100% MUT required for infectivity, couldn't the authors passage their libraries once and measure the % composition change.

We now reference Fig. S18 and its discussion in the text early in the g/P section. To differentiate between a mixture and a 100% g/P linkage, we thought about this idea. However, we preferred to assemble separately T7-WT and T7-mutant in different tubes and then express the phages separately in TXTL, dilute the TXTL reactions, and then mix the TXTL reactions (called equivolume in the main) for several reasons:

- The main reason is that ensuring a single amplification of the phage mixture is technically challenging and would require anti-T7 antibodies.
- Allowing multiple phage cycles of the mixture on *E. coli* B would add more bias than cell-free mix, such as the WT propagating faster compared to the mutated one. The overall phage titers would be different. Keeping all cell-free makes more comparable settings.

Overall, we estimated that doing this experiment in cell-free only was preferable.

12. Which is what: ($1.9 \pm 0.4 \times 10^6$, $1.7 \pm 0.4 \times 10^7$ PFU/mL, respectively).

This was clarified in the text (p. 8, Line 264):

‘($1.9 \pm 0.4 \times 10^6$ for equimolar, $1.7 \pm 0.4 \times 10^7$ PFU/mL for equivolume)’

13. How were these conditions different than the overnight. i.e., how long is extended period and what was the LPS concentration for the 50% mixture: Separately assembled T7-vWT phages were poorly susceptible to ReLPS when incubated with 0.4 mg/mL ReLPS for an extended period at 37 C (10-fold decrease of titer from $\sim 6 \times 10^8$ PFU/mL to $\sim 5 \times 10^7$ PFU/mL).

This was poor phrasing originally. Extended period is indeed corresponding to overnight. This was changed. In this work, we used either:

- “mild conditions”: 0.2 mg/mL, 2 h, 37°C.
- “Stringent conditions”: 0.4 mg/mL, overnight, 37°C.

“Stringent conditions” were used here to demonstrate the lack of sensibility of T7WT to ReLPS and at the same time enable complete ReLPS-sensitive phage ejection from a phage mixture. This was added to the methods (p. 18 line 651):

‘Two conditions were used in this work: mild conditions (0.2 mg/mL, 2 h, 37°C), stringent conditions (0.4 mg/mL, overnight, 37°C).’

14. what % is this and was this accounted for in the calculations: A few undefined plaques were also detected on rfaC from the equal volume mix of pre-assembled T7-vWT and -rfaD-1 phages after the ReLPS ejection and amplification on WT host.

This was accounted for in the calculations. In fact, we estimated the ratio of “newly formed mutants” (0.00022 ± 0.00013) from the equal-volume mix and retrieved it from the ratio of (mut/WT)/(wt/WT) (0.029 ± 0.022) obtained on the equimolar mix titered on *E. coli* B and rfaC. This contribution was found to be negligible yielding a final ratio (mut/WT)/(wt/WT) value of 0.029 ± 0.022 .

We added in parentheses (p. 8 Line 273):

‘(negligible percentage compared to co-expression)’

15. Can the authors introduce and describe the (i) and (ii) in better detail. Seem abruptly introduced.

We added more description p. 8 line 283.

These two equations describe the assumption that the ratio of pure p/G phages (mut/MUT and wt/WT) as well as opposing p/G (mut/WT and wt/MUT) on all the total number of phages that carry a mut or a wt genotype are the same.

We renamed these equations (1) and (2). We added the following sentence after the equations (1) and (2):

‘Equations (1) and (2) describe the assumption that the ratio of pure p/G phages (mut/MUT and wt/WT) as well as opposing p/G (mut/WT and wt/MUT) on all the total number of phages that carry a mut or a wt genotype are the same.’

16. What is ‘pure and opposing p/Gs’?

A pure p/G describes a phage that displays only tail fibers that correspond to its genomic sequence, for example, wt genotype and only WT tail fibers. An opposing p/G phage displays only tail fibers that do not correspond to the genomic sequence for example wt genotype and only MUT tail fibers.

We added the following sentences p. 8 line 285:

‘We define pure p/G as a phage that displays only tail fibers that correspond to its genomic sequence, for example, wt genotype and only WT tail fibers. Conversely, we define opposing p/G a phage that displays only tail fibers that do not correspond to the genomic sequence for example wt genotype and only MUT tail fibers.’

17. Can the authors please elaborate why H2 was chosen for the hypothetical plot for random mix in Figure 4e or 4f?. The authors begin the discussion assuming only one monomer is required for phage infection. Isn't that H0? Also, would be interesting to model if 1 monomers, 2 monomers, 3 monomers... are required for infectivity. We calculated the theoretical fully randomized relative abundance of the six different g/P hybrids under the most permissive assumption, namely that the integration of at least one S541R mutant monomer into one of the six trimeric tail fibers of a given phage is necessary and sufficient to infect rfaC ReLPS (Fig. 4d).

Note that there is no figure 4f, just figure 4e after revisions. We actually did not choose H2, but H0. In the legend of figure 4e, the random is H0. We added this to the main text and to the figure 4e legend.

18. How does the % composition change if the authors grow the 50% mixture for a single infection cycle on E. coli B and then plaque on E. coli B and ReLPS, essentially converting all the

mut/Mix to mut/MUT and wt/Mix to wt/WT? This experiment should differentiate whether one tail fiber of MUT is needed or if 100% MUT is needed. (same comment as 11b).

We thank the reviewer for this idea. We did not perform this experiment as we could not achieve a single phage cycle for the equimolar mix.

19. Please include DOI: 10.1016/j.cell.2019.09.015 here: 'The tip of the tail fiber gene 17 (amino acids 472-554), determinant of the phage's host range^{53 54 48}, can be exchanged between phages or mutated to adapt to new hosts^{55 56 57 58}.'

We added this reference p. 10 line 325.

20. Please specific in the figure what library each phage came from:

Twelve clonal phages from each of the three hosts (Fig. 5d-f) were purified, propagated in their respective host, phenotyped on all hosts (Fig. S22, S23, S24) and sequenced (Fig. S25). 6 plaques were picked from T7-E1 (6 first from the top in Fig. 5d, now labeled 1 to 6), 3 plaques from T7-E2 (three following in Fig. 5d, labeled 7 to 9), and 3 plaques from T7-E3 (last three in Fig. 5d, labeled 10 to 12) from rfaC, lpcA and rfaE. This order was kept in Fig. S23 and S24. In Fig. S25, the same procedure as for Fig. S5 was performed for rfaD while for rfaG, single plaques could only be picked for T7-E3. This likely explains why many more mutations (silent or not) are observed in the tail fiber sequences of rfaG variants.

We updated accordingly Fig. 5d, Fig. S23 and Fig. S24.

21. The authors should consider including this data in main text as a bar graph. Just the C, E, A mutants:

This strongly supports a g/P linkage in PHEIGES. Interestingly, titers were somewhat lower on the rfaC as compared to the rfaE or lpcA host strains although they are believed to share the same ReLPS serotype. This may suggest different membrane properties between the knockouts.

This statement was changed as the difference between the different phenotypes was minimal and our data do not support major differences, which is also supported by the literature (<https://www.nature.com/articles/s41598-019-46100-3>). We removed the sentence 'Interestingly, titers were somewhat lower on the rfaC than the rfaE or lpcA host strains although they are believed to share the same ReLPS serotype. This may suggest different membrane properties between the knockouts.'

22. Discussion is severely lacking. Please expand the discussion to describe how this library generation approach compares in efficiency of expanding phage host range to other methods and the nuances between each.

On the mutant scale level (libraries), our approach can be only compared to the work done by Huss and coworkers (ORACLE). In Oracle, a complete systematic residue substitution

on the 80 last amino acids of the tail fiber is done, without a selection process. PHEIGES selects the mutations that improve the host range, based on a library generated by mutagenic PCR. PHEIGES, however, can do this for larger fragments of the tail fiber, twice larger in our work. Doing this with ORACLE would add a considerable amount of work, but none with PHEIGES. Thus, the two approaches are somewhat complementary in generating phages with improved host ranges. Yet, PHEIGES presents several advantages.

We added the following paragraph at the end of the section ‘Selection for T7 phage host range expansion via PHEIGES g/P linkage’ (p12-13 lines 399-426):

‘PHEIGES presents several advantages compared with ORACLE²⁰ and other current methods²²^{23 16 62} in generating and identifying phages with improved host ranges. Firstly, the library approach to generate infectious phages can be performed on any large genetic parts of the phage within a day as compared to ORACLE that necessitates additional cloning and downstream selection steps. Secondly, PHEIGES has the capability to address multiple amino acid mutation per sequence by manipulating the mutagenic PCR parameters. Thirdly, PHEIGES provides greater versatility through its all-cell-free method. In ORACLE, tail fiber residues are classified based on the relative abundance each variant when the library is propagated into the host. While probably correct as a first approach, other naturally occurring mutations in the tail fiber as well as the tail complex are omitted. The phage variants in higher abundance detected in ORACLE were likely to also carry mutations, for instance, in the tail genes as reported here (Table S5) and in other works^{52 49}. Analyzing these other mutations for each variant (1660 variants) would have been too cumbersome and sequence extensive. With PHEIGES, we show that while other mutations appear in the tail genes, the tail fiber mutations are the major determinants of the host range. By re-assembling phages from the selected variants with improved host range with tail fiber-only mutations and phages with tail-only mutations, we show that the tail fiber-only mutations are sufficient to infect *ReLPS* strains, whereas mutations in the tail genes are not. Fourthly, PHEIGES is compatible with phage library assembly with or without cellular amplification. For the exploration of gain of function variants, cellular amplification is dispensable given the high phage titers attained by PHEIGES (>10¹⁰ PFU/mL). Alternatively, if a total p/G is desired, cellular amplification of the TXTL phage library remains an option. Fifthly, PHEIGES addresses the common challenge encountered in editing methods like CRISPR, where there is often an overwhelming prevalence of wild-type phage compared to variants. With PHEIGES, the absence of fragments results in the absence of phages. Consequently, a library of fragments leads to a library of phages with the relative abundance of each phage variant directly proportional to the relative concentration of DNA fragments. Finally, we anticipate that PHEIGES could enable radical phage hybridization and genome reshuffling, overcoming limitations seen in conventional co-infection.’

23. If g/P weren’t coupled, couldn’t one simply add a single passage step, to make all the libraries mut/mut? Could the authors please include this in the discussion, and potential biases of this approach?

We added the following paragraph in the discussion section (p15 lines 491-513):

‘PHEIGES enables synthesizing and selecting T7 phages with engineered tail fiber because a significant proportion of their phenotypes are linked to their genotypes in batch mode TXTL reactions (p/G linkage). Without such a linkage (e.g., capsid protein and other phages yet to be

tested), one could explore a library of PHEIGES-engineered phages by selecting through a single passage *in vivo*. This method, however, has potential biases. A single pass *in vivo*, for instance, would kinetically bias the selected phages to the infection process (e.g., binding kinetics, binding strength). Without an *in vivo* step, if the library is not p/G linked all the solutions are represented. In that case, the issue is how to cope with the false positives. Future works will focus on studying the nature of p/G linkage in TXTL and how to improve it. Regardless, PHEIGES is compatible with *in vivo* passages, either after DNA assembly by transformation or after cell-free synthesis, that could or could not be used depending on the downstream application. If the goal is to produce non-infectious phages for biotechnology applications, such as vaccine scaffolds (ref), it is not necessary to pass *in vivo*. If the goal is to produce infectious phages, passing *in vivo* can be done but does not necessarily present advantages. PHEIGES already maximizes the exploration of mutations with a mutation rate that can be tuned at will during the PCR amplification. Phages are selected at the last step, thus minimizing *in vivo* biases. Because the cell-free synthesis is not compartmentalized, PHEIGES produces a myriad of chimeric phages at the structure level. These steps are efficient and show minimal constraints compared to the propagation in living cells (e.g., difficult-to-perform phage genome transformation followed by living cell amplifications with various anti-phage systems and metabolisms). Lastly, PHEIGES enables the synthesis of some non-coli phages. FelixO1, for instance, is a salmonella phage requiring a BLS2 laboratory. PHEIGES enables engineering and synthesizing phages in BSL1 conditions accessible by all laboratories. As such, PHEIGES extends accessibility to phage engineering.’

REVIEWERS' COMMENTS

Reviewer #1 (Remarks to the Author):

The authors have addressed most of my concerns.

They did not, however, address the concern #1 in my review. I did not claim that there is no novelty in the findings of linkage between the phenotype and genotype. I claimed that this novel linkage did not lead to novel findings, but rather to known mutations of LPS-binding receptor.

In light of the other corrections, I tend to agree that pursuing these experiments will require at least 6 months, and therefore may be out of the scope of the current manuscript.

Reviewer #2 (Remarks to the Author):

The authors have adequately addressed the comments raised in my previous review. Just two comments:

(1) Figure S1: A positive control is located bottom left, not bottom right.

(2) Figure S7_legend: The authors didn't address my last comment.

Reviewer #3 (Remarks to the Author):

The authors have addressed all my concerns. The modified text is much improved. The added experimental details and expanded discussion were much appreciated.

Response to Reviewers - Revision 2

REVIEWERS' COMMENTS

Reviewer #1 (Remarks to the Author):

The authors have addressed most of my concerns.

They did not, however, address the concern #1 in my review. I did not claim that there is no novelty in the findings of linkage between the phenotype and genotype. I claimed that this novel linkage did not lead to novel findings, but rather to known mutations of LPS-binding receptor. In light of the other corrections, I tend to agree that pursuing these experiments will require at least 6 months, and therefore may be out of the scope of the current manuscript.

We thank the reviewer for his review which allowed us to improve the manuscript.

Reviewer #2 (Remarks to the Author):

The authors have adequately addressed the comments raised in my previous review. Just two comments:

(1) Figure S1: A positive control is located bottom left, not bottom right.

Corrected.

(2) Figure S7_legend: The authors didn't address my last comment.

The typo was corrected (the number of DNA fragments was changed to 6 and 5).

Thank you for your insightful feedback throughout the reviewing process.

Reviewer #3 (Remarks to the Author):

The authors have addressed all my concerns. The modified text is much improved. The added experimental details and expanded discussion were much appreciated.

Thank you for your insightful feedback throughout the reviewing process.